# Gumiho: A Hybrid Architecture
# to Prioritize Early Tokens in Speculative Decoding

**Jinze Li** [1 2]  **Yixing Xu** [1]  **Haiduo Huang** [1 3]  **Xuanwu Yin** [1]  **Dong Li** [1]  **Edith C.H. Ngai** [* 2]  **Emad Barsoum** [1]

lijinze-hku@connect.hku.hk, huanghd@stu.xjtu.edu.cn, chngai@eee.hku.hk
{yixing.xu, Xuanwu.Yin, d.li, emad.barsoum}@amd.com

## Abstract

Speculative decoding (SPD) aims to accelerate the auto-regressive token generation process of a target Large Language Model (LLM). Some approaches employ a draft model with multiple heads to predict a sequence of future tokens, where each head handles a token in the sequence. The target LLM verifies the predicted sequence and accepts aligned tokens, enabling efficient multi-token generation. However, existing methods assume that all tokens within a sequence are equally important, employing identical head structures and relying on a single-generation paradigm, either serial or parallel. To this end, we theoretically demonstrate that initial tokens in the draft sequence are more important than later ones. Building on this insight, we propose Gumiho, a hybrid model combining serial and parallel heads. Specifically, given the critical importance of early tokens, we employ a sophisticated Transformer architecture for the early draft heads in a serial configuration to improve accuracy. For later tokens, we utilize multiple lightweight MLP heads operating in parallel to enhance efficiency. By allocating more advanced model structures and longer running times to the early heads, Gumiho achieves improved overall performance. The experimental results demonstrate that our method outperforms existing approaches, fully validating its effectiveness. Our code is available at https://github.com/AMD-AIG-AIMA/Gumiho.

[1]Advanced Micro Devices, Inc., Beijing, China [2]Department of Electrical and Electronic Engineering, The University of Hong Kong [3]Institute of Artificial Intelligence and Robotics, Xi'an Jiaotong University. Correspondence to: Edith C.H. Ngai <chngai@eee.hku.hk>.

*Proceedings of the 42^{nd} International Conference on Machine Learning*, Vancouver, Canada. PMLR 267, 2025. Copyright 2025 by the author(s).

## 1. Introduction

While Large Language Models (LLMs) (Achiam et al., 2023; Touvron et al., 2023) have demonstrated impressive capabilities, their auto-regressive inference introduces significant latency challenges, especially as the number of parameters continues to increase. Speculative decoding (SPD) (Leviathan et al., 2023; Chen et al., 2023) has emerged as a promising solution to this problem. It leverages smaller models to efficiently propose draft tokens for future steps, which are then verified in parallel by the LLM. Specifically, in each draft round, the draft model generates a sequence of multiple draft tokens, and the target LLM verifies these tokens in parallel. This generation-verification process constitutes a draft round. These draft tokens are accepted only if they match the LLM's original output. If a mismatch occurs, starting from the first divergent token, all subsequent draft tokens are discarded.

Recent advances (Cai et al., 2024; Li et al., 2024a;b; Ankner et al., 2024) have shown that smaller models, which leverage the hidden states of the LLM itself, can achieve substantial speedups in inference while maintaining output quality. Medusa (Cai et al., 2024) predicts multiple future tokens in parallel using MLPs with the last verified token's hidden state, while Hydra (Ankner et al., 2024) and Eagle (Li et al., 2024a) generate tokens serially.

Although Medusa's parallel prediction paradigm runs faster through simultaneous draft token predictions, it relies solely on hidden states from previously verified tokens, making it blind to earlier unverified predictions within the current draft round. Serial methods like Eagle, Eagle-2 and Hydra can fully utilize previously generated draft tokens, but their sequential paradigm tends to slow down the drafting process, making it less efficient. More critically, all approaches treat all tokens within a draft round as equally important, which is unsuitable for SPD. **In our view, the tokens generated earlier in each draft round hold more importance than those generated later.** This is because when the first incorrect token is encountered, it causes both that token and all subsequent draft tokens to be discarded, even if the following tokens are correct. We will formally present a

theorem and mathematically prove that prioritizing the initial tokens of a sequence consistently improves the mean accepted tokens ($\tau$) in each draft round.

Motivated by this, we propose Gumiho, a hybrid model that prioritizes the initial positions in the draft sequence, combining both sequential and parallel architectures. We employ a serial structure comprising a two-layer Transformer, enabling comprehensive modeling of token dependencies and context for early tokens. The subsequent tokens, which are relatively less critical, are predicted in parallel through simple MLPs, thereby enhancing computational efficiency. By combining the serial and parallel architectures, our hybrid design achieves higher acceptance lengths and less processing time in each draft-verification cycle at the same time. The key idea of our hybrid model lies in two folds: (1) it allocates more parameters and leverages serial processing for crucial early token predictions, maximizing accuracy where it matters most, and (2) it employs efficient parallel computation with simple architecture for later tokens, reducing the overall computational cost.

In addition, we enhance Eagle-2's dynamic tree mechanism by introducing Full Tree Attention (FTA). While Eagle-2's dynamic tree selectively expands promising nodes and re-ranks draft tokens for optimal verification, this re-ranking process can result in shorter candidate sequences when later tokens are discarded due to low scores, potentially reducing the mean accepted tokens ($\tau$). We observed that tokens generated by parallel heads exhibit correlations, as they share both input and purpose. Specifically, each $n$-th head is designed to predict the token that should appear $n$ positions away from the input token. Inspired by this, our full tree attention mechanism supplements shorter paths with tokens from longer paths at corresponding positions, thereby increasing the mean accepted tokens ($\tau$) of shorter candidate paths. Since these supplementary tokens come from existing longer candidates, their $q$, $k$, and $v$ have already been computed, incurring no additional computational overhead.

Our contributions are summarized as follows:

- We propose Gumiho, a hybrid structure model for SPD, inspired by the observation that earlier tokens have more impact on the overall sequence length accepted, while later tokens are relatively less critical. We prioritize the generation of earlier tokens by allocating more computational resources and using a serial approach to enhance accuracy, while simpler models are employed in parallel for the later tokens to improve computational efficiency.

- We demonstrate through theoretical analysis that tokens appearing earlier in the draft sequence have a more significant impact on the overall accepted length.

- We propose a full tree attention mechanism for tree candidates, allowing tokens from longer candidates to augment shorter ones. This approach further increases the acceptance length without incurring additional computational overhead.

- We conduct comprehensive experiments to demonstrate Gumiho's superior performance compared to existing methods.

## 2. Related Works

With the widespread adoption of large language models (LLMs), significant research has been devoted to accelerating their inference through techniques such as distillation (Hinton, 2015; Bercovich et al., 2024; Zhao et al., 2024; Fu et al., 2024), low-bit quantization (Hubara et al., 2018; Shen et al., 2020; Kim et al., 2021; Zadeh et al., 2020; Zafrir et al., 2019), pruning (Gale et al., 2019; Sanh et al., 2020; Kurtic et al., 2022; Voita et al., 2019), and innovative network architecture designs (Gu & Dao, 2023; Wu et al., 2020). These approaches aim to reduce the computational cost of each forward pass to improve efficiency. However, they often involve a trade-off (Donisch et al., 2024), as these optimizations can partially compromise model performance, requiring a balance between generation quality and computational overhead. Speculative decoding, a draft-then-verify paradigm (Xia et al., 2023), achieves lossless acceleration by leveraging the original LLM for verification.

Drafting approaches can be broadly categorized into two paradigms (Xia et al., 2024): independent drafting which employs draft models that can be deployed without additional training, and self-drafting which requires dedicated training processes to develop effective draft models.

Independent drafting typically uses a separate, smaller model to generate multiple future tokens concurrently, thereby enhancing the efficiency of speculative decoding. For example, using T5-small to accelerate T5-XXL (Leviathan et al., 2023). These off-the-shelf drafters do not need extra training or architectural modifications and benefit from the inherent alignment in prediction behaviors due to shared tokenizers and pretraining processes. However, independent drafting requires additional work to find or train a compatible model that matches the target LLM. This becomes more challenging when smaller versions of the LLM don't exist.

Orthogonal to independent drafting, self-drafting typically uses the target LLM itself (Liu et al., 2024a; Du et al., 2024; Elhoushi et al., 2024; Gloeckle et al., 2024; Cai et al., 2024; Li et al., 2024a; Zimmer et al., 2024; Xiao et al., 2024; Zhang et al., 2024; Brown et al., 2024; Liu et al., 2024b), utilizing features like its hidden states for more efficient drafting. Medusa (Cai et al., 2024) is one of the studies to leverage the hidden state of the original LLM as input for

draft models. It employs $K$ distinct MLPs as draft models, each predicting one of $K$ future tokens. Since these $K$ draft models operate independently, they can execute in parallel, enabling Medusa to generate $K$ tokens in a single forward pass. Hydra (Ankner et al., 2024) is a sequential variant of Medusa, transforming the parallel MLPs into a serial architecture. It feeds the unverified tokens from draft models as input to subsequent ones. This sequential approach enables each draft model to leverage more contextual information when predicting later tokens, enhancing the quality of successive predictions. Eagle (Li et al., 2024a) advances the architecture by converting the serial MLPs into serial Transformers and introducing concatenated token-hidden state pairs as input. Eagle-2 (Li et al., 2024b) further innovates by implementing a dynamic tree candidate selection mechanism to enhance token prediction efficiency.

Our method falls within the self-drafting category. Different from previous self-drafting methods, we utilize the fact that the importance of a token decreases as its position moves back and propose different architectures for front-positioned tokens and later ones.

## 3. Method

In this section, we begin by introducing the preliminaries of speculative decoding (SPD). We then present a theorem and provide a rigorous mathematical proof to validate that tokens at the beginning of the sequence are more crucial than those at the end. Finally, we propose Gumiho, a novel method derived from our theorem.

### 3.1. Preliminaries

**LLM Decoding**   The process of generating text from LLMs is termed decoding: the sequential production of tokens in response to an input prompt. This generation process follows an auto-regressive pattern, where each token $y_t$ is generated by sampling from a probability distribution conditioned on both the initial prompt $z$ and all previously generated tokens $y_{<t}$. In practice, the key-value cache ($kv_{<t}$) of previously generated tokens is maintained, with the model taking both $kv_{<t}$ and the current token $y_t$ as inputs for efficient generation.

Let $\mathcal{M}_L$ denote the Large Language Model, which comprises two components: the decoder layer $f_L(\cdot)$ and $LM\_head$ that maps the embeddings back to the vocabulary with size $|\mathcal{V}|$. The vanilla decoding process can be formulated as:

$$kv_{<t+1}, h_{t+1} = f_L(kv_{<t}, e(y_t)),$$
$$y_{t+1} \sim \text{Softmax}(LM\_head(h_{t+1})), \quad (1)$$

where $h_{t+1}$ denotes the hidden states of the final decoder layer, and $e(\cdot)$ is an embedding function that maps tokens to their corresponding vector representations. In the following,

we define $y_{t+1} = \mathcal{M}_L(y_t) \triangleq \mathcal{M}_L(y_t, kv_{<t})$ and ignore $kv_{<t}$ for simplicity.

**Speculative Decoding**   Auto-regressive text generation in LLMs is time-consuming. Speculative decoding addresses this limitation by employing a smaller, faster draft model $\mathcal{M}_S$ to generate candidate tokens ahead of time. These candidate tokens, commonly referred to as drafts, are then verified in parallel by the target LLM $\mathcal{M}_L$ using rejection sampling (Leviathan et al., 2023), *i.e.*, if any token in the draft sequence is rejected, all subsequent tokens are discarded, and the draft-verification process resumes from the last accepted token.

During each draft-verification iteration, $\mathcal{M}_S$ generates a sequence of $D$ draft tokens $\{\hat{y}_{t+i}\}_{i=1}^{D}$. Subsequently, $\mathcal{M}_L$ verifies these drafts in parallel according to Eq.(1):

$$y_{t+1} \sim \mathcal{M}_L(y_t),$$
$$y_{t+2} \sim \mathcal{M}_L(\hat{y}_{t+1}), \quad (2)$$
$$\vdots$$
$$y_{t+D} \sim \mathcal{M}_L(\hat{y}_{t+D-1}).$$

The number of accepted draft tokens for each iteration is determined by comparing the draft sequence $\{\hat{y}_{t+1}, ..., \hat{y}_{t+D}\}$ with the verified sequence $\{y_{t+1}, ..., y_{t+D}\}$ using rejection sampling.

**Metrics**   We assess the method's performance by measuring its acceleration effect. Specifically, we use the speedup ratio as a metric, which is calculated by dividing the speed of our proposed method by the speed of standard (vanilla) decoding:

$$\text{Speedup ratio} = \frac{\text{speed}_{\text{Gumiho}}}{\text{speed}_{\text{vanilla}}} \quad (3)$$

The speed of each method is calculated by dividing the total number of generated tokens by the total processing time:

$$\begin{aligned} \text{speed} &= \frac{\text{total tokens}}{\text{total time}} \\ &= \frac{\text{mean accepted tokens} \times \text{draft rounds}}{\text{average time} \times \text{draft rounds}} \\ &= \frac{\text{mean accepted tokens}}{\text{average time}} \quad (4) \end{aligned}$$

Following existing works (Li et al., 2024b; Ankner et al., 2024), we primarily use *mean accepted tokens* ($\tau$) as the main metric. We also present the differences in draft time across different methods to demonstrate the effectiveness of our approach in improving efficiency.

### 3.2. Theoretical Analysis

In this section, we prove that tokens at the beginning of the draft sequence are more crucial than those at the end.

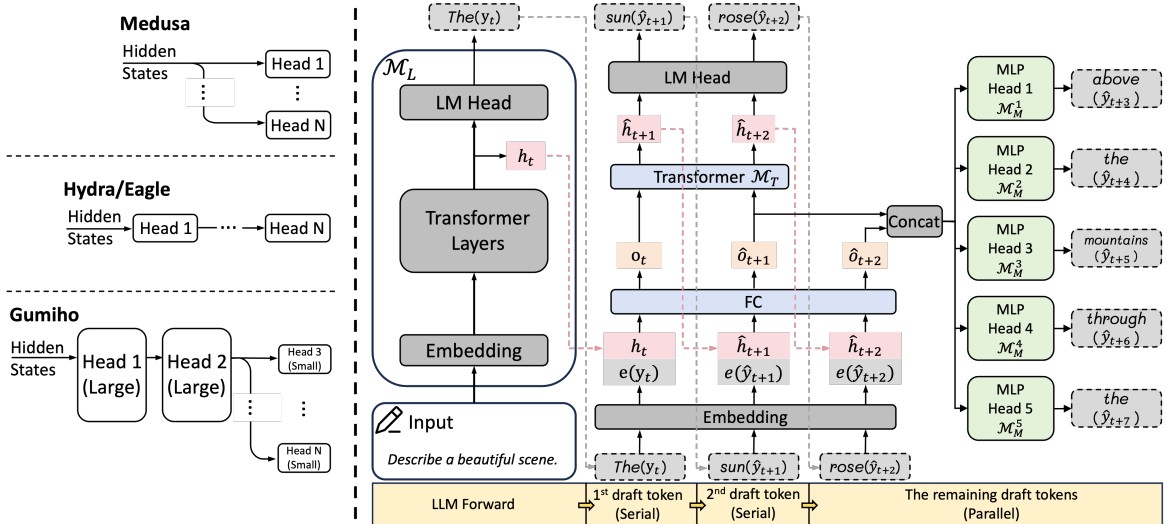

*Figure 1.* **Left**: Differences between our proposed Gumiho and existing methods: Unlike existing approaches that use similar models to predict every token in a sequence, we propose that initial tokens are more critical than later ones. So we employ a larger model with a serial structure to generate the early tokens, while leveraging smaller parallel models for the later ones. **Right**: Overview of Gumiho. Given an LLM input *Describe a beautiful scene.*, Gumiho predicts the next 7 draft tokens (*sun rose above the mountains through the*). The first two tokens (*sun* and *rose*) are deemed critical and are produced sequentially using the Transformer $\mathcal{M}_T$ for higher accuracy. The remaining tokens are generated simultaneously through the MLP heads, optimizing for computational efficiency.

Consider a draft model that predicts 3 tokens at a time with a uniform acceptance probability of $0.8$ at each position. The expected length $\mathbb{E}[L]_{\text{original}}$ of accepted tokens per draft round is:

$$\mathbb{E}[L]_{\text{original}} = 1 \times P(L=1) + 2 \times P(L=2) + 3 \times P(L=3)$$
$$= \sum_{i=1}^{3} P(L \geq i)$$
$$= 0.8 + 0.8^2 + 0.8^3 = 1.95,$$

where $L$ represents the accepted length determined by the target LLM's verification after each draft round. $\mathbb{E}[\cdot]$ denotes the expectation operator, and $P(L=i)$ represents the probability that the accepted length $L$ is equal to $i$.

Now, suppose we redistribute the model's parameters to prioritize earlier positions, *i.e.*, allocating more parameters to predict the first position and fewer for subsequent positions. Assume this improved structure creates position-dependent acceptance probabilities of $0.85$, $0.8$, and $0.75$ for the first, second, and third positions respectively. The expected length $\mathbb{E}[L]_{\text{improved}}$ becomes:

$$\mathbb{E}[L]_{\text{improved}} = \sum_{i=1}^{3} P(L \geq i)$$
$$= 0.85 + 0.85 \times 0.8 + 0.85 \times 0.8 \times 0.75 = 2.04.$$

This example empirically demonstrates that when the overall token accuracy remains constant, improving the accuracy of the initial token can increase the mean accepted tokens ($\tau$).

In the following, we provide a theorem to generalize the above example to a broader scenario. Given a draft sequence with length $D$, we first have an original setting with acceptance probabilities $\{p_i\}_{i=1}^{D}$ and denote $\mathbb{E}[L]_{\text{original}}$ as its mean accepted tokens ($\tau$). Since errors accumulate when predicting a sequence, the acceptance probability of later tokens tends to be lower than that of earlier tokens in practical scenarios. Based on this observation, we have:

$$1 \geq p_1 \geq p_2 \geq \cdots \geq p_D \geq 0. \tag{5}$$

Then, we define an improved setting whose sequence is separated by index $d$ with $1 < d < D$. In this setting, acceptance probabilities $\tilde{p}_i$ are modified as follows:

$$\tilde{p}_i = \begin{cases} p_i + \zeta_i, & 1 \leq i \leq d \\ p_i - \zeta_i, & d < i \leq D \end{cases},$$
$$s.t.\ 0 \leq \{\zeta_i\}_{i=1}^{D} \leq 1, \quad 0 \leq \{\tilde{p}_i\}_{i=1}^{D} \leq 1,$$
$$\sum_{i=1}^{d} \zeta_i = \sum_{j=d+1}^{D} \zeta_j, \quad \zeta_i < p_i. \tag{6}$$

In this improved setting, we increase the acceptance probabilities for the first $d$ tokens by a small amount $\zeta_i$ and decrease those for the remaining tokens by the same total amount. In this way, the sum of the acceptance probabilities remains unchanged. We denote the mean accepted tokens ($\tau$) in this improved setting as $\mathbb{E}[L]_{\text{improved}}$. With these definitions above, we can derive the following theorem:

**Theorem 3.1.** *The mean accepted tokens ($\tau$) under the improved probability distribution exceeds that of the original distribution:*

$$\mathbb{E}[L]_{\text{improved}} \geq \mathbb{E}[L]_{\text{original}}.$$

This theorem shows that redistributing the acceptance probabilities across sequence tokens by increasing the accuracies of the initial tokens and decreasing those of the later tokens can improve the overall expected performance. A detailed proof of this theorem is provided in Appendix A.

### 3.3. Gumiho

Inspired by Theorem 3.1, we propose Gumiho, a model that prioritizes the front part of the draft sequence. Gumiho consists of two main components: heads for generating draft tokens and a full tree attention mechanism for verification.

**Gumiho Heads.** As shown in Fig. 1, Gumiho introduces a hybrid architecture that distinguishes itself from existing methods. Unlike approaches that rely solely on a single serial or parallel structure and employ uniform head size across all positions, Gumiho combines large serial heads with small parallel heads to enhance accuracy and efficiency.

The serial component aims to increase the accuracy of the initial tokens and comprises a two-layer Transformer $\mathcal{M}_{\text{T}}$ that predicts initial draft tokens sequentially. Specifically, $\mathcal{M}_{\text{T}}$ generates the first two tokens of the draft sequence autoregressively. Similar to Eagle-2, our method concatenates the hidden state $h_t \in \mathbb{R}^d$ with the corresponding output token embedding $e(y_t) \in \mathbb{R}^d$ generated by LLM $\mathcal{M}_L$ at time step $t$, and employs a fully connected layer FC to reduce the dimension from $2d$ to $d$:

$$o_t = \text{FC}(\text{cat}(e(y_t), h_t)), \tag{7}$$

where $\text{cat}$ denotes the concatenation operation, and $o_t \in \mathbb{R}^d$. Then, the concatenated result $o_t$ is fed into the serial component $\mathcal{M}_{\text{T}}$, which sequentially generates the first two drafts:

$$\hat{h}_{t+1} = \mathcal{M}_{\text{T}}(o_t), \quad \hat{o}_{t+1} = \text{FC}(\text{cat}(e(\hat{y}_{t+1}), \hat{h}_{t+1})), \tag{8}$$

$$\hat{h}_{t+2} = \mathcal{M}_{\text{T}}(\hat{o}_{t+1}), \quad \hat{o}_{t+2} = \text{FC}(\text{cat}(e(\hat{y}_{t+2}), \hat{h}_{t+2})). \tag{9}$$

For simplicity, we omit the input and output of KV cache in $\mathcal{M}_{\text{T}}$, and also the step of using Softmax to obtain $\hat{y}_{t+1}$, $\hat{y}_{t+2}$, which is similar to Eq. (1).

The parallel component aims to speed up the generation of the remaining tokens while maintaining accuracy and consists of five different MLPs $\{\mathcal{M}_{\text{M}}^i\}_{i=1}^5$ running concurrently. These MLPs share the same architecture, consisting of two fully connected (FC) layers with a ReLU activation function in between. They also share the same input, *i.e.*, $\text{cat}(\hat{o}_{t+1}, \hat{o}_{t+2})$ which concatenate the two outputs generated by the serial model $\mathcal{M}_{\text{T}}$. The outputs of MLPs represent the draft tokens at the following five positions:

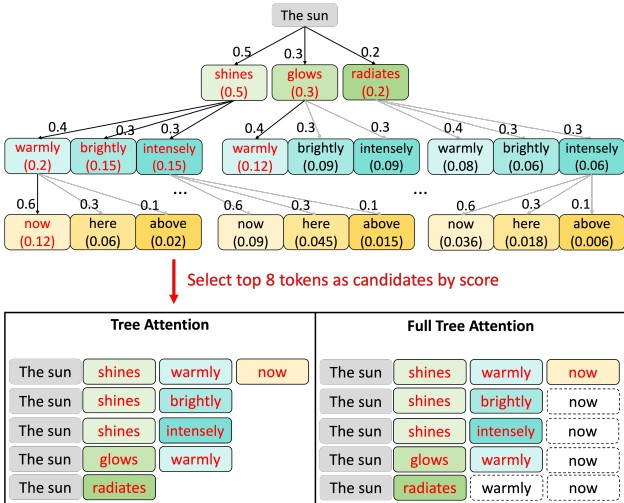

*Figure 2.* Our proposed Full Tree Attention enhances shorter candidate paths by borrowing tokens from other tree nodes, thereby increasing the likelihood that candidates achieve longer acceptance lengths. Note that for each depth in the tree, we only have $s = 3$ different tokens.

$$\hat{h}_{t+2+i} = \mathcal{M}_{\text{M}}^i(\text{cat}(\hat{o}_{t+1}, \hat{o}_{t+2})), \quad i = 1, ..., 5. \tag{10}$$

Given the hidden states $\{\hat{h}_{t+i}\}_{i=1}^7$, we obtain the draft tokens $\{\hat{y}_{t+i}\}_{i=1}^7$ using Eq. (1).

**Full Tree Attention (FTA).** A key distinction between our Gumiho and Eagle-2 lies in the use of parallel heads for generating subsequent tokens. To fully leverage this parallel paradigm, we introduce the full tree attention mechanism, which enhances the existing Tree Attention mechanism employed by Eagle-2.

In Eagle-2, tokens at each position are generated in an autoregressive manner, meaning that each subsequent token is entirely dependent on the tokens generated in the previous. Conversely, our parallel heads $\mathcal{M}_{\text{M}}^i$ generate tokens for all positions simultaneously. This parallel generation paradigm removes dependencies between these tokens, as they are determined solely by the outputs of the preceding serial outputs generated by $\mathcal{M}_{\text{T}}$. The independence between tokens enables us to perform a full traversal connection operation on the tokens generated in parallel. Specifically, any two tokens generated by two different $\mathcal{M}_{\text{M}}^i$ can be connected to form a candidate path. As illustrated in Fig. 2, after the serial heads output the tokens *the* and *sun*, the three parallel heads generate $s$ subsequent tokens for each position ($s = 3$ in Fig. 2). These $s$ tokens at each position can be arbitrarily combined with tokens from other positions, resulting in a total of $s^3$ candidate paths with only $3s$ different tokens by the time we complete the third MLP head. In Fig. 2, the score for each token is displayed, calculated by multiplying the score of the preceding token with the confidence of the

current MLP in generating specific tokens.

To select candidate paths for verification, we choose the top eight tokens with the highest scores. As shown in Fig. 2, traditional tree attention often results in some candidate paths being very short because tokens on early positions usually have higher scores. To address this issue, our proposed FTA mechanism supplements shorter paths with tokens from corresponding positions in longer paths. This is reasonable since any two tokens from different positions among the parallel-generated tokens can be combined, and the borrowed tokens from longer candidate paths do not conflict with the original tokens in shorter paths. This ensures the coherence of the final candidate paths, maintaining the integrity of the generated sequence. Consequently, this approach increases the average length of candidate paths, enhancing the overall performance of the model.

It is worth noting that FTA incurs no additional computational overhead, as the borrowed tokens already exist in other candidate paths within the tree, with their query, key, and value computations already completed. In the original attention tree, these tokens were simply discarded, while our approach unblocks them and allows shorter candidates to access and utilize them. The rejection sampling method (Leviathan et al., 2023) for token selection ensures that the appending of borrowed tokens does not impact previously selected tokens. Consequently, after applying FTA, each candidate's acceptance length equals or exceeds that of the original implementation.

## 4. Experiments

In this section, we compare our proposed Gumiho with other SOTA methods to show the priority of our approach. Then, we conduct several ablation studies to validate the effectiveness of each part of our method.

### 4.1. Experimental Setup

We conduct experiments using seven target LLMs: Vicuna-7B/13B (Chiang et al., 2023), Llama2-chat-7B/13B/70B (Touvron et al., 2023), and Llama3-instruct-8B/70B (Meta, 2024). Target LLMs are fixed during training, with only the draft heads being trained. Following Eagle and Eagle-2 (Li et al., 2024a;b), we train our draft model on the ShareGPT dataset. Our Gumiho model comprises a Transformer model and five MLPs to predict the next seven draft tokens: the Transformer autoregressively generates the first two tokens, and the remaining five are predicted in parallel by the MLPs. Training details and hyperparameters can be found in Appendix C.

We evaluate the performance across multiple benchmarks: MT-Bench (Zheng et al., 2023) for multi-turn dialogue, HumanEval (Chen et al., 2021) for code generation,

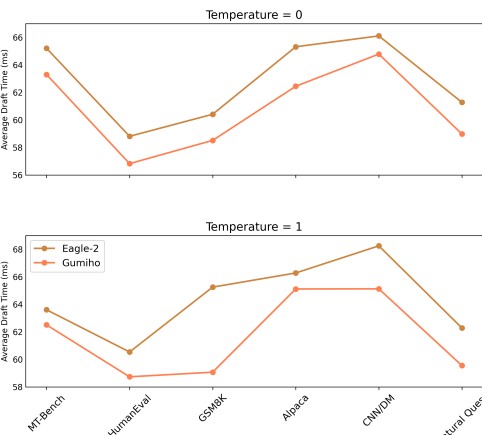

*Figure 3.* Comparison of average draft time (lower is better) with different temperatures. Both results are based on Vicuna 7B.

GSM8K (Cobbe et al., 2021) for mathematical reasoning, Alpaca (Taori et al., 2023) for general instruction-following, CNN/Daily Mail (Nallapati et al., 2016) for summarization, and Natural Questions (Kwiatkowski et al., 2019) for question answering. We conduct model training using 8×AMD Instinct MI250 GPUs. For evaluation, we use a single MI250 GPU for all models except the 70B variant, which requires 4×MI250 GPUs due to its larger size. Additionally, we include evaluation results using a single NVIDIA A100 GPU in Appendix B.

We compare our method against several existing approaches: Medusa (Cai et al., 2024) with multiple parallel MLP heads, Hydra (Ankner et al., 2024) with sequential MLP heads, Eagle (Li et al., 2024a) and Eagle-2 (Li et al., 2024b) with sequential single-layer Transformer head. Eagle-2 shares the same model parameters as Eagle but distinguishes itself by incorporating a dynamic tree to generate candidates.

In line with Eagle-2 (Li et al., 2024b), we also conduct experiments with temperature settings of 0 and 1. A temperature of 0 means that the target LLM uses a greedy sampling method, where the token with the highest probability is selected at each position. In contrast, a temperature of 1 increases the diversity of the output by applying post-processing to the logits at the current position, rather than directly selecting the token with the highest probability. When the temperature is set to 1, Eagle-2 excludes methods like Medusa since their relaxed acceptance criteria under non-greedy sampling do not ensure lossless acceleration. We follow their experimental settings in this work.

### 4.2. Performance Comparison

Experimental results are shown in Tab. 1. The *Speedup* metric quantifies the actual end-to-end acceleration ratio of token generation speed compared to vanilla auto-regressive generation, while $\tau$ is the average number of tokens accepted

*Table 1.* Speedup ratios and mean accepted tokens ($\tau$) of different methods. V represents Vicuna, L2 represents LLaMA2-Chat, and L3 represents LLaMA3-Instruct. We present the results of different methods across six datasets. *Mean* represents the average performance across these six datasets.

| Model | Method | MT-Bench | | HumanEval | | GSM8K | | Alpaca | | CNN/DM | | Natural Ques. | | Mean | |
|---|---|---|---|---|---|---|---|---|---|---|---|---|---|---|---|
| | | Speedup | $\tau$ | Speedup | $\tau$ | Speedup | $\tau$ | Speedup | $\tau$ | Speedup | $\tau$ | Speedup | $\tau$ | Speedup | $\tau$ |
| | | | | | | | | Temperature=0 | | | | | | | |
| V 7B | Medusa | 1.96× | 2.50 | 2.15× | 2.69 | 2.01× | 2.59 | 1.94× | 2.48 | 1.60× | 2.02 | 1.68× | 2.05 | 1.89× | 2.39 |
| | Hydra | 2.47× | 3.59 | 2.65× | 3.78 | 2.49× | 3.67 | 2.44× | 3.58 | 1.92× | 2.70 | 2.01× | 2.86 | 2.33× | 3.36 |
| | Eagle | 2.61× | 3.82 | 2.96× | 4.20 | 2.67× | 4.00 | 2.41× | 3.66 | 2.35× | 3.34 | 2.10× | 3.13 | 2.52× | 3.69 |
| | Eagle-2 | 2.88× | 5.00 | 3.27× | 5.35 | 2.93× | 4.94 | 2.71× | 4.85 | 2.45× | 4.11 | 2.24× | 3.84 | 2.74× | 4.68 |
| | Gumiho(ours) | **3.15×** | **5.29** | **3.65×** | **5.77** | **3.10×** | **5.06** | **2.83×** | **4.87** | **2.73×** | **4.48** | **2.34×** | **3.88** | **2.97×** | **4.89** |
| V 13B | Medusa | 2.03× | 2.58 | 2.24× | 2.77 | 2.08× | 2.64 | 2.04× | 2.44 | 1.67× | 2.10 | 1.70× | 2.10 | 1.96× | 2.44 |
| | Hydra | 2.65× | 3.65 | 2.88× | 3.86 | 2.69× | 3.67 | 2.65× | 3.49 | 2.08× | 2.82 | 2.16× | 2.86 | 2.52× | 3.39 |
| | Eagle | 2.87× | 3.90 | 3.25× | 4.29 | 2.88× | 3.90 | 2.64× | 3.50 | 2.58× | 3.49 | 2.21× | 2.92 | 2.74× | 3.66 |
| | Eagle-2 | 3.16× | 4.93 | 3.68× | 5.42 | 3.19× | 4.82 | 3.01× | 4.89 | 2.79× | 4.27 | 2.41× | 3.69 | 3.04× | 4.67 |
| | Gumiho(ours) | **3.36×** | **5.16** | **4.11×** | **5.97** | **3.39×** | **5.04** | **3.07×** | **4.88** | **2.91×** | **4.41** | **2.52×** | **3.76** | **3.23×** | **4.87** |
| L2 7B | Eagle | 2.22× | 3.00 | 2.53× | 3.58 | 2.21× | 3.09 | 2.04× | 2.88 | 2.08× | 2.78 | 1.88× | 2.64 | 2.16× | 3.00 |
| | Eagle-2 | 2.91× | 4.76 | 3.30× | 5.38 | 2.87× | 4.76 | 2.81× | 4.65 | 2.53× | 4.10 | 2.52× | 4.16 | 2.82× | 4.64 |
| | Gumiho(ours) | **3.07×** | **4.90** | **3.55×** | **5.60** | **3.00×** | **4.81** | **2.85×** | **4.55** | **2.66×** | **4.18** | **2.59×** | **4.16** | **2.95×** | **4.70** |
| L2 13B | Eagle | 2.59× | 3.30 | 2.96× | 3.90 | 2.61× | 3.45 | 2.41× | 3.16 | 2.39× | 3.09 | 2.15× | 2.82 | 2.52× | 3.29 |
| | Eagle-2 | 3.17× | 4.76 | 3.78× | 5.53 | 3.23× | 4.88 | 3.03× | 4.62 | 2.84× | 4.27 | 2.76× | 4.12 | 3.13× | 4.70 |
| | Gumiho(ours) | **3.34×** | **4.98** | **4.05×** | **5.87** | **3.35×** | **5.02** | **3.12×** | **4.66** | **2.93×** | **4.40** | **2.84×** | **4.20** | **3.27×** | **4.85** |
| L2 70B | Eagle-2 | 2.51× | 4.52 | 2.98× | 5.24 | 2.63× | 4.63 | 2.48× | 4.42 | 2.04× | 3.72 | 2.14× | 3.88 | 2.47× | 4.40 |
| | Gumiho(ours) | **2.83×** | **4.71** | **3.35×** | **5.43** | **2.90×** | **4.69** | **2.70×** | **4.46** | **2.37×** | **4.08** | **2.35×** | **3.90** | **2.76×** | **4.54** |
| L3 8B | Eagle-2 | 2.16× | 4.36 | 2.51× | 5.06 | 2.22× | 4.45 | 2.25× | 4.88 | 1.82× | 3.81 | 1.75× | 3.54 | 2.12× | 4.35 |
| | Gumiho(ours) | **2.38×** | **4.48** | **2.77×** | **5.18** | **2.49×** | **4.63** | **2.44×** | **4.88** | **2.00×** | **3.94** | **1.93×** | **3.64** | **2.34×** | **4.46** |
| L3 70B | Eagle-2 | 2.94× | 4.17 | 3.65× | 5.09 | 3.17× | 4.34 | 3.12× | **4.74** | 2.54× | 3.66 | 2.48× | 3.50 | 2.98× | 4.25 |
| | Gumiho(ours) | **3.38×** | **4.28** | **4.28×** | **5.25** | **3.79×** | **4.58** | **3.48×** | 4.58 | **2.91×** | **3.80** | **2.87×** | **3.59** | **3.45×** | **4.35** |
| | | | | | | | | Temperature=1 | | | | | | | |
| V 7B | Eagle-2 | 2.51× | 4.30 | 2.67× | 4.52 | 2.46× | 4.47 | 2.38× | 4.37 | 2.15× | 3.70 | 2.02× | 3.50 | 2.37× | 4.16 |
| | Gumiho(ours) | **2.61×** | **4.42** | **2.84×** | **4.62** | **2.73×** | **4.52** | **2.46×** | **4.40** | **2.38×** | **3.94** | **2.10×** | **3.51** | **2.52×** | **4.23** |
| V 13B | Eagle-2 | 2.81× | 4.37 | 3.32× | 4.96 | 2.80× | 4.43 | 2.66× | 4.46 | 2.51× | 3.92 | 2.25× | 3.50 | 2.73× | 4.27 |
| | Gumiho(ours) | **2.93×** | **4.54** | **3.55×** | **5.30** | **2.84×** | **4.59** | **2.77×** | **4.54** | **2.58×** | **4.04** | **2.36×** | **3.72** | **2.84×** | **4.46** |
| L2 7B | Eagle-2 | 2.66× | 4.63 | 2.95× | 5.15 | 2.70× | **4.76** | 2.52× | 4.40 | 2.34× | 3.98 | 2.29× | 4.02 | 2.58× | 4.49 |
| | Gumiho(ours) | **2.79×** | **4.64** | **3.19×** | **5.27** | **2.78×** | 4.67 | **2.64×** | 4.40 | **2.47×** | **4.05** | **2.44×** | **4.08** | **2.72×** | **4.52** |
| L2 13B | Eagle-2 | 3.01× | 4.60 | 3.58× | 5.34 | 3.09× | 4.76 | 2.91× | 4.49 | 2.71× | 4.15 | 2.66× | 4.08 | 2.99× | 4.57 |
| | Gumiho(ours) | **3.18×** | **4.82** | **3.86×** | **5.71** | **3.24×** | **4.94** | **2.98×** | **4.62** | **2.80×** | **4.28** | **2.76×** | **4.16** | **3.14×** | **4.75** |
| L2 70B | Eagle-2 | 2.28× | 4.41 | 2.73× | 5.15 | 2.42× | 4.59 | 2.31× | 4.30 | 1.87× | 3.67 | 2.00× | 3.72 | 2.27× | 4.30 |
| | Gumiho(ours) | **2.60×** | **4.65** | **3.15×** | **5.46** | **2.66×** | **4.61** | **2.50×** | **4.43** | **2.15×** | **3.98** | **2.22×** | **3.95** | **2.55×** | **4.51** |
| L3 8B | Eagle-2 | 1.93× | 4.04 | 2.32× | 4.80 | 2.06× | 4.27 | 2.03× | **4.57** | 1.67× | 3.55 | 1.59× | 3.27 | 1.93× | 4.08 |
| | Gumiho(ours) | **2.13×** | **4.14** | **2.55×** | **4.95** | **2.29×** | **4.42** | **2.19×** | 4.55 | **1.86×** | **3.64** | **1.72×** | **3.32** | **2.12×** | **4.17** |
| L3 70B | Eagle-2 | 2.85× | 4.07 | 3.57× | 4.97 | 3.13× | 4.31 | 3.00× | **4.65** | 2.47× | 3.58 | 2.42× | 3.45 | 2.91× | 4.17 |
| | Gumiho(ours) | **3.29×** | **4.20** | **4.20×** | **5.17** | **3.69×** | **4.49** | **3.34×** | 4.43 | **2.84×** | **3.71** | **2.85×** | **3.57** | **3.37×** | **4.26** |

by the target LLM per draft round after verification.

Across diverse target LLMs, model sizes, and temperature settings, Gumiho demonstrates superior performance. Overall, Gumiho surpasses the existing SOTA method EAGLE-2 by 4.5%~15.8%. The performance gains are particularly pronounced with 70B model variants. At temperature 0, Gumiho outperforms EAGLE-2 by 11.7% on LLaMA2 70B and 15.8% on LLaMA3 70B. This substantial improvement is primarily attributed to enhancements in $\tau$ and a reduction in draft time. Specifically, the output hidden state for 70B models has a dimension of 8192, compared to 4096 in the 7B and 13B models. While the larger hidden state increases computational complexity, it also amplifies the benefits of our model parallelization, significantly reducing drafting time and further boosting the speedup ratio.

In Fig. 3, we present the time required by different models for a draft round. From Eq. (4), it is evident that a shorter draft time leads to a higher speedup ratio, resulting in improved performance. Fig. 3 demonstrates that the draft time of Gumiho is consistently shorter than that of Eagle-2 across all datasets regarding different temperatures, highlighting

the superiority of our approach. This is primarily attributed to the efficiency of the parallel MLP heads in Gumiho.

### 4.3. Ablation Studies

In this section, we evaluate the contribution of each component in our method to the overall performance. Specifically, we conduct ablation studies focusing on three key aspects: (1) The depth of the serial head (Transformer head), where we vary the number of Transformer layers to assess its impact; (2) The width of parallel heads (MLP heads), where we experiment with different numbers of MLP heads; and (3) the effectiveness of full tree attention (FTA). These ablation studies aim to provide a deeper understanding of the architectural design choices in our proposed method and their respective contributions to the final performance. We also present a detailed comparison of draft head accuracy in Appendix D. Unless otherwise specified, the ablation experiments are conducted using Vicuna 7B as the target LLM, with MT-Bench as the test dataset and the temperature set to 0.

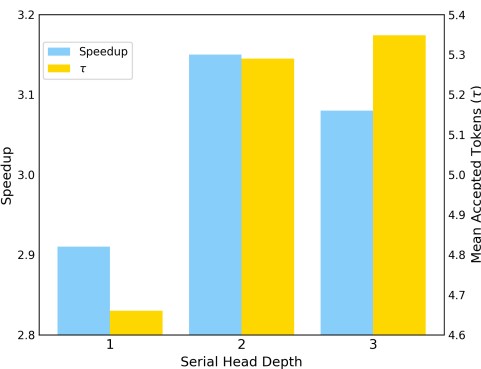

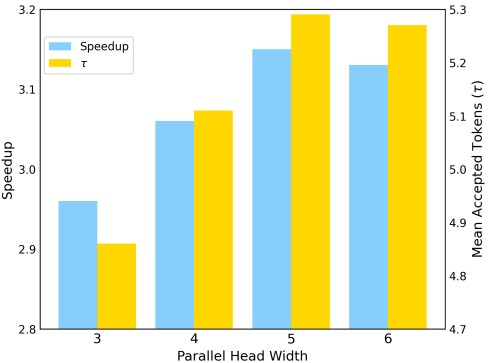

*Figure 4.* Ablation study on serial head depth. The serial head is a Transformer model, whose depth represents the number of layers within the Transformer architecture.

*Figure 5.* Ablation study on parallel head width. Parallel heads refer to the MLPs in Gumiho, and the width indicates the number of MLP models.

**Serial head depth.** The serial head depth refers to the number of layers in the Transformer model. This Transformer serves as the initial head responsible for generating the first two tokens in a draft sequence. In this study, we vary the number of layers in the Transformer model to examine how the depth of the initial head affects the model's overall performance. The experimental results shown in Fig. 4 reveal that reducing the number of Transformer layers results in a decline in $\tau$, which underscores the significant impact of the initial heads. However, when the depth increases from 2 to 3 layers, although $\tau$ improves further, the speedup ratio decreases. This is because the overall speedup depends not only on $\tau$ but also on the time required to complete a single draft process. Using a three-layer Transformer substantially increases the drafting time, which ultimately reduces the speedup effect.

It is worth noting that using varied depths to generate the first two tokens, such as a two-layer Transformer for the first token and a one-layer Transformer for the second, may seem intuitive but proves inefficient during inference. Since Transformers with different architectures cannot share their key-value (KV) caches, each head must compute its cache independently. This prevents cache reuse between heads, increasing the computational overhead. Our approach employs identical serial heads throughout the model, only trains one single Transformer model, and reuses it for auto-regressive token generation during inference. This architectural uniformity enables efficient KV cache sharing across the entire generation process.

**Parallel Head Width.** Parallel head width refers to the number of MLP heads in Gumiho, which run in parallel to generate subsequent tokens in the draft sequence. The experimental results are shown in Fig. 5. Note that increasing the number of MLP heads initially improves performance but eventually leads to a decline. This is because increasing the number of parallel heads enhances the model's capacity to predict longer draft sequences, thereby improving its ability

to generate more accepted tokens. Additionally, since MLP heads operate in parallel, increasing their number does not significantly increase runtime. A higher number of mean accepted tokens with a similar runtime leads to an improved performance. However, this improvement does not scale indefinitely. All MLP heads share the same input embedding, which is derived from the concatenation of outputs from the preceding Transformer head. During training, this shared embedding serves as the input to every MLP head and is simultaneously shaped by the back-propagated losses from all of them. As the number of MLP heads increases, the embedding must encode a growing amount of information to meet the requirements of each additional head. This leads to excessive information being compressed into the limited embedding space, which reduces the clarity and specificity of the information available to each MLP head and ultimately causes performance degradation.

**Effectiveness of Full Tree Attention (FTA).** We verify the effectiveness of this component by conducting experiments with or without using FTA in our model. The experimental results are presented in Tab. 2. They indicate that removing FTA impacts the mean accepted tokens and diminishes the speedup effect.

**Wall Clock Time of Different Components.** To provide a granular understanding of the model's performance, we report the wall clock time of key components in the pipeline, as shown in Tab. 3.

*Table 2.* Ablation study on the FTA mechanism.

|  | Speedup | $\tau$ |
|---|---|---|
| w/o Full Tree Attention | 3.10× | 5.18 |
| w/ Full Tree Attention | 3.15× | 5.29 |

_Table 3._ Ablation study on the wall time.

| Components | Wall Time (ms) |
| --- | --- |
| 1st Serial Head | 2.80 |
| 2nd Serial Head | 3.46 |
| Parallel Head | 2.02 |
| Full Tree Attention | 3.41 |
| Other Computation | 6.11 |
| Verification | 45.50 |

## 5. Conclusion

The core idea of this paper is to rigorously prove that the accuracy of early tokens in a draft sequence is more critical than that of later ones in speculative decoding. In other words, given a fixed budget for model parameter size and overall execution time, prioritizing the heads responsible for generating the initial tokens can improve overall performance. Building on this insight, we propose Gumiho, a novel approach that employs a hybrid head design. Specifically, Gumiho allocates a larger proportion of the parameter and execution time budget to the head responsible for generating the initial tokens. This head is a Transformer model with a serial structure designed for these initial tokens, ensuring higher accuracy. For those heads that generate later tokens, Gumiho employs lightweight MLPs and parallelizes their execution to produce multiple tokens simultaneously. This hybrid design achieves improved performance by balancing accuracy and efficiency: the serial Transformer enhances the accuracy of the initial tokens, while the parallel MLPs reduce overall generation time. Experimental results validate the effectiveness of our approach, demonstrating that Gumiho surpasses existing state-of-the-art methods.

## 6. Limitation

Our proposed method, while achieving significant speedup, utilizes a more parameter-heavy draft model compared to architectures like Eagle and Medusa. Specifically, the incorporation of a two-layer Transformer head alongside five parallel MLP heads, trained concurrently, results in increased GPU memory consumption during the training phase.

## Impact Statement

This paper presents work whose goal is to advance the field of Machine Learning. There are many potential societal consequences of our work, none which we feel must be specifically highlighted here.

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

# A. Detailed Proof of Theorem 3.1

Given $\{p_i\}_{i=1}^D$ and $\{\tilde{p}_i\}_{i=1}^D$ defined in Eq. (5) and Eq. (6) in the main paper, the mean accepted tokens ($\tau$) for the original and improved settings are expressed as:

$$\mathbb{E}[L]_{\text{original}} = \sum_{k=1}^{D} P_{ori}(L \geq k) = \sum_{k=1}^{D}(\prod_{i=1}^{k} p_i), \tag{11}$$

and

$$\mathbb{E}[L]_{\text{improved}} = \sum_{k=1}^{D} P_{imp}(L \geq k) = \sum_{k=1}^{D}(\prod_{i=1}^{k} \tilde{p}_i). \tag{12}$$

We aim to prove that:

$$\mathbb{E}[L]_{\text{improved}} \geq \mathbb{E}[L]_{\text{original}}. \tag{13}$$

We introduce an auxiliary probability sequence $P_i'$ that concentrates the scattered changes $\zeta_i$ at two adjacent positions $d$ and $d+1$. Specifically, we define:

$$P_i' = \begin{cases} p_i + \zeta, & i = d \\ p_i - \zeta, & i = d+1 \\ p_i, & otherwise, \end{cases} \quad s.t. \quad \zeta = \sum_{i=1}^{d} \zeta_i = \sum_{j=d+1}^{D} \zeta_j. \tag{14}$$

Here, we assume

$$p_d + \zeta \leq 1. \tag{15}$$

We will discuss the cases of $p_d + \zeta > 1$ at the end.

With these assumptions hold, the corresponding mean accepted tokens ($\tau$) for this concentrated setting is:

$$\mathbb{E}[L]_{\text{concentrate}} = \sum_{k=1}^{D} P_{con}(L \geq k) = \sum_{k=1}^{D}(\prod_{i=1}^{k} P_i'). \tag{16}$$

In the following, we will prove that:

$$\mathbb{E}[L]_{\text{concentrate}} \geq \mathbb{E}[L]_{\text{original}}, \tag{17}$$
$$\mathbb{E}[L]_{\text{improved}} \geq \mathbb{E}[L]_{\text{concentrate}}. \tag{18}$$

Before proceeding with the main proof, let us examine the special case where $p_{d+1} = 1$. In this case, the ordering constraint $1 \geq p_1 \geq p_2 \geq \cdots \geq p_d \geq p_{d+1}$ implies that all probabilities are equal: $p_1 = p_2 = \cdots = p_d = p_{d+1} = 1$. Given that $p_i + \zeta_i \leq 1$ for all $i = \{1, 2, \cdots, d\}$, we must have $\zeta_1 = \zeta_2 = \cdots = \zeta_d = 0$. This leads to $\zeta = \sum_{i=1}^{d} \zeta_i = \sum_{i=d+1}^{D} \zeta_i = 0$, meaning that $\{p_i\}_{i=1}^D$ and $\{\tilde{p}_i\}_{i=1}^D$ are exactly the same. As a result,, $\mathbb{E}[L]_{\text{improved}} = \mathbb{E}[L]_{\text{original}}$.

For the remainder of the proof, we assume $p_{d+1} < 1$.

**A.1. The proof of $\mathbb{E}[L]_{\text{concentrate}} \geq \mathbb{E}[L]_{\text{original}}$**

Define

$$\Delta E = \mathbb{E}[L]_{\text{concentrate}} - \mathbb{E}[L]_{\text{original}} = \sum_{k=1}^{D} \Delta E_k, \tag{19}$$

where $\Delta E_k = \prod_{i=1}^{k} P_i' - \prod_{i=1}^{k} p_i$ represents the contributions to the difference of the expected value for each position $k$.

**Step 1: Analyze of $\Delta E$.**

We separate $\Delta E$ into three parts, where $\Delta E_1 = \sum_{k=1}^{d} \Delta E_k$, $\Delta E_2 = \Delta E_{d+1}$ and $\Delta E_3 = \sum_{k=d+2}^{D} \Delta E_k$.

For $1 \le k < d$, $P_k' = p_k$ and $\Delta E_k = 0$. Therefore:

$$\begin{aligned}
\Delta E_1 &= \Delta E_d \\
&= \prod_{i=1}^{d} P_i' - \prod_{i=1}^{d} p_i \\
&= \prod_{i=1}^{d-1} p_i \cdot (p_d + \zeta) - \prod_{i=1}^{d} p_i \\
&= \zeta \prod_{i=1}^{d-1} p_i.
\end{aligned} \tag{20}$$

For $k = d + 1$, we have:

$$\begin{aligned}
\Delta E_2 &= \Delta E_{d+1} \\
&= \prod_{i=1}^{d+1} P_i' - \prod_{i=1}^{d+1} p_i \\
&= \prod_{i=1}^{d-1} p_i \cdot (p_d + \zeta)(p_{d+1} - \zeta) - \prod_{i=1}^{d+1} p_i \\
&= \prod_{i=1}^{d-1} p_i \cdot (\zeta p_{d+1} - \zeta p_d - \zeta^2) \\
&= \zeta(p_{d+1} - p_d - \zeta) \prod_{i=1}^{d-1} p_i.
\end{aligned} \tag{21}$$

For $k > d + 1$, the difference arises from the terms $(p_d + \zeta)$ and $(p_{d+1} - \zeta)$ in the product. Thus:

$$\begin{aligned}
\Delta E_3 &= \sum_{k=d+2}^{D} \Delta E_k \\
&= \sum_{k=d+2}^{D} \left( \prod_{i=1}^{k} P_i' - \prod_{i=1}^{k} p_i \right) \\
&= \sum_{k=d+2}^{D} \left[ (\prod_{i=1}^{d-1} p_i)(p_d + \zeta)(p_{d+1} - \zeta)(\prod_{i=d+2}^{k} p_i) - \prod_{i=1}^{k} p_i \right] \\
&= \sum_{k=d+2}^{D} \left[ (\prod_{i=1}^{d-1} p_i)\zeta(p_{d+1} - p_d - \zeta)(\prod_{i=d+2}^{k} p_i) \right].
\end{aligned} \tag{22}$$

By combining Eq. (20) $\sim$ Eq. (22), we have:

$$\begin{aligned}
\Delta E &= \Delta E_1 + \Delta E_2 + \Delta E_3 \\
&= \zeta(p_{d+1} - p_d + 1 - \zeta) \prod_{i=1}^{d-1} p_i + \sum_{k=d+2}^{D} \left[ (\prod_{i=1}^{d-1} p_i)\zeta(p_{d+1} - p_d - \zeta)(\prod_{i=d+2}^{k} p_i) \right].
\end{aligned} \tag{23}$$

**Step 2: Scaling.**

Define $A = \prod_{i=1}^{d-1} p_i \zeta$ and recall that in Eq. (5) we have $1 \geq p_1 \geq p_2 \geq \cdots \geq p_D \geq 0$, then the total difference $\Delta E$ becomes:

$$
\begin{aligned}
\Delta E &= A(p_{d+1} - p_d + 1 - \zeta) + \sum_{k=d+2}^{D} \left[ A(p_{d+1} - p_d - \zeta)(\prod_{i=d+2}^{k} p_i) \right] \\
&= A \left[ 1 + (p_{d+1} - p_d - \zeta) + (p_{d+1} - p_d - \zeta) \sum_{k=d+2}^{D} \prod_{i=d+2}^{k} p_i \right] \\
&= A \left[ 1 - (p_d - p_{d+1} + \zeta) - (p_d - p_{d+1} + \zeta) \sum_{k=d+2}^{D} \prod_{i=d+2}^{k} p_i \right] \\
&\geq A \left[ 1 - (p_d - p_{d+1} + \zeta) - (p_d - p_{d+1} + \zeta) \sum_{k=d+2}^{D} p_{d+1}^{k-d-1} \right] \\
&= A \left[ 1 - (p_d - p_{d+1} + \zeta)(1 + \sum_{k=d+2}^{D} p_{d+1}^{k-d-1}) \right].
\end{aligned}
\tag{24}
$$

**Step 3: Simplifying the sum.**

Note that the last term $1 + \sum_{k=d+2}^{D} p_{d+1}^{k-d-1}$ in Eq. (24) is a geometric series. Therefore, we have:

$$
\begin{aligned}
1 + \sum_{k=d+2}^{D} p_{d+1}^{k-d-1} &= 1 + p_{d+1} + p_{d+1}^2 + p_{d+1}^3 + \cdots + p_{d+1}^{D-d-1} \\
&= \frac{1 - p_{d+1}^{D-d}}{1 - p_{d+1}}.
\end{aligned}
\tag{25}
$$

Substitute this back into $\Delta E$, and we have:

$$
\Delta E \geq A \left[ 1 - (p_d - p_{d+1} + \zeta)(\frac{1 - p_{d+1}^{D-d}}{1 - p_{d+1}}) \right].
\tag{26}
$$

**Step 4: Proving $\Delta E \geq 0$**

To ensure $\Delta E \geq 0$, we need:

$$
\begin{aligned}
&(p_d - p_{d+1} + \zeta)(\frac{1 - p_{d+1}^{D-d}}{1 - p_{d+1}}) \leq 1, \\
\iff &(p_d - p_{d+1} + \zeta)(1 - p_{d+1}^{D-d}) \leq 1 - p_{d+1}, \\
\iff &(p_d + \zeta) - p_{d+1}^{D-d}(p_d - p_{d+1} + \zeta) \leq 1.
\end{aligned}
\tag{27}
$$

Since $1 \geq p_d \geq p_{d+1} \geq 0$ and $\zeta \geq 0$, it follows that $p_{d+1}^{D-d}(p_d - p_{d+1} + \zeta) \geq 0$. Besides, since we have $p_d + \zeta \leq 1$ in Eq. (15), the inequality holds. Thus, $\Delta E \geq 0$. Implying:

$$
\mathbb{E}[L]_{\text{concentrate}} \geq \mathbb{E}[L]_{\text{original}}.
\tag{28}
$$

**A.2. Proving that $\mathbb{E}[L]_{\text{improved}} \geq \mathbb{E}[L]_{\text{concentrate}}$**

We introduce a series of auxiliary sequences with the goal of proving the following inequality chain. Note that the first line and the last line in the following inequality chain represent the improved setting and the concentrate setting, respectively.

Note that only two elements are different in every two consecutive lines (except for the last two lines). Specifically, we merge $\zeta_i$ into $\zeta_d$ one at a time for $1 \le i \le d - 1$.

$$
\mathbb{E}(p_1 + \zeta_1, \quad p_2 + \zeta_2, \quad p_3 + \zeta_3, \quad \cdots, \quad p_d + \zeta_d, \quad p_{d+1} - \zeta_{d+1}, \quad p_{d+2} - \zeta_{d+2}, \quad \cdots, \quad p_D - \zeta_D)
$$

$$
\ge
$$

$$
\mathbb{E}(p_1, \quad p_2 + \zeta_2, \quad p_3 + \zeta_3, \quad \cdots, \quad p_d + \zeta_d + \zeta_1, \quad p_{d+1} - \zeta_{d+1}, \quad p_{d+2} - \zeta_{d+2}, \quad \cdots, \quad p_D - \zeta_D)
$$

$$
\ge
$$

$$
\mathbb{E}(p_1, \quad p_2, \quad p_3 + \zeta_3, \quad \cdots, \quad p_d + \zeta_d + \zeta_1 + \zeta_2, \quad p_{d+1} - \zeta_{d+1}, \quad p_{d+2} - \zeta_{d+2}, \quad \cdots, \quad p_D - \zeta_D)
$$

$$
\ge
$$

$$
\cdots
$$

$$
\ge
$$

$$
\mathbb{E}(p_1, \quad p_2, \quad p_3, \quad \cdots, \quad p_d + \sum_{i=1}^{d} \zeta_i, \quad p_{d+1} - \zeta_{d+1}, \quad p_{d+2} - \zeta_{d+2}, \quad \cdots, \quad p_D - \zeta_D)
$$

$$
\ge
$$

$$
\mathbb{E}(p_1, \quad p_2, \quad p_3, \quad \cdots, \quad p_d + \sum_{i=1}^{d} \zeta_i, \quad p_{d+1} - \sum_{i=d+1}^{D} \zeta_i, \quad p_{d+2}, \quad \cdots, \quad p_D)
$$

$$(29)$$

**First Part of the Inequality Chain**

We begin by proving the first inequality in the chain, which involves transferring $\zeta_1$ from $p_1$ to $p_d$:

$$
\mathbb{E}(p_1 + \zeta_1, \quad p_2 + \zeta_2, \quad p_3 + \zeta_3, \quad \cdots, \quad p_d + \zeta_d, \quad p_{d+1} - \zeta_{d+1}, \quad p_{d+2} - \zeta_{d+2}, \quad \cdots, \quad p_D - \zeta_D)
$$

$$
\ge
$$

$$
\mathbb{E}(p_1, \quad p_2 + \zeta_2, \quad p_3 + \zeta_3, \quad \cdots, \quad p_d + \zeta_d + \zeta_1, \quad p_{d+1} - \zeta_{d+1}, \quad p_{d+2} - \zeta_{d+2}, \quad \cdots, \quad p_D - \zeta_D)
$$

$$(30)$$

Let $\Delta E_4^1$ denote the difference between the left-hand side (LHS) and the right-hand side (RHS) of the above inequality:

$$
\begin{aligned}
\Delta E_4^1 =& \mathbb{E}(p_1 + \zeta_1, p_2 + \zeta_2, p_3 + \zeta_3, \cdots, p_d + \zeta_d, p_{d+1} - \zeta_{d+1}, p_{d+2} - \zeta_{d+2}, \cdots, p_D - \zeta_D) \\
& - \mathbb{E}(p_1, p_2 + \zeta_2, p_3 + \zeta_3, \cdots, p_d + \zeta_d + \zeta_1, p_{d+1} - \zeta_{d+1}, p_{d+2} - \zeta_{d+2}, \cdots, p_D - \zeta_D) \\
=& (p1 + \zeta_1 - p_1) + \sum_{k=2}^{d-1} \left[ \prod_{i=1}^{k}(p_i + \zeta_i) - p_1 \prod_{i=2}^{k}(p_i + \zeta_i) \right] + \\
& \left[ \prod_{i=1}^{d}(p_i + \zeta_i) - p_1 \left( \prod_{i=2}^{d-1}(p_i + \zeta_i) \right)(p_d + \zeta_1 + \zeta_d) \right] \left[ 1 + \sum_{k=d+1}^{D} ( \prod_{i=d+1}^{k}(p_i - \zeta_i) \right] \\
\geq& \zeta_1 + \left[ \prod_{i=1}^{d}(p_i + \zeta_i) - p_1 \left( \prod_{i=2}^{d-1}(p_i + \zeta_i) \right)(p_d + \zeta_1 + \zeta_d) \right] \left[ 1 + \sum_{k=d+1}^{D} ( \prod_{i=d+1}^{k}(p_i - \zeta_i) \right] \\
\geq& \zeta_1 + \prod_{i=2}^{d-1}(p_i + \zeta_i) \left[ (p_1 + \zeta_1)(p_d + \zeta_d) - p_1(p_d + \zeta_1 + \zeta_d) \right] \left[ 1 + \sum_{k=d+1}^{D} ( \prod_{i=d+1}^{k}(p_i - \zeta_i) \right] \\
=& \zeta_1 + \prod_{i=2}^{d-1}(p_i + \zeta_i)\zeta_1 (p_d + \zeta_d - p_1) \left[ 1 + \sum_{k=d+1}^{D} ( \prod_{i=d+1}^{k}(p_i - \zeta_i) \right]
\end{aligned}
\tag{31}
$$

If $p_d + \zeta_d - p_1 \geq 0$, then $\Delta E_4^1 \geq 0$. This directly implies that the first inequality holds.

If $p_d + \zeta_d - p_1 < 0$, then

$$
\begin{aligned}
\Delta E_4^1 \geq& \zeta_1 - \prod_{i=2}^{d-1}(p_i + \zeta_i)\zeta_1 (p_1 - p_d - \zeta_d) \left[ 1 + \sum_{k=d+1}^{D} (p_{d+1}^{k-d}) \right] \\
=& \zeta_1 - \prod_{i=2}^{d-1}(p_i + \zeta_i)\zeta_1 (p_1 - p_d - \zeta_d) \frac{1 - p_{d+1}^{D-d+1}}{1 - p_{d+1}} \\
\geq& \zeta_1 - \zeta_1 (p_1 - p_d - \zeta_d) \frac{1}{1 - p_{d+1}} \\
=& \zeta_1 \left[ 1 - (p_1 - p_d - \zeta_d) \frac{1}{1 - p_{d+1}} \right] \\
=& \zeta_1 \left[ \frac{(1 - p_{d+1}) - (p_1 - p_d - \zeta_d)}{1 - p_{d+1}} \right] \\
=& \zeta_1 \left[ \frac{(1 - p_1) + (p_d + \zeta_d - p_{d+1})}{1 - p_{d+1}} \right]
\end{aligned}
\tag{32}
$$

Given that $p_1 \leq 1$ and $p_{d+1} \leq p_d < 1$, the numerator $(1 - p_1) + (p_d + \zeta_d - p_{d+1})$ and the denominator $1 - p_{d+1}$ are both positive. Hence, $\Delta E_4^1 \geq 0$.

In both cases, $\Delta E_4^1 \geq 0$, thereby proving Inequality 30.

Similarly, consider further transferring $\zeta_2$ from $p_2$ to $p_d$:

$$
\begin{array}{l}
\mathbb{E}(p_1, \quad p_2 + \zeta_2, \quad p_3 + \zeta_3, \quad \cdots, \quad p_d + \zeta_d + \zeta_1, \quad p_{d+1} - \zeta_{d+1}, \quad p_{d+2} - \zeta_{d+2}, \quad \cdots, \quad p_D - \zeta_D) \\
\\
\qquad\qquad\qquad\qquad\qquad\qquad\qquad \geq \\
\\
\mathbb{E}(p_1, \quad p_2, \quad p_3 + \zeta_3, \quad \cdots, \quad p_d + \zeta_d + \zeta_1 + \zeta_2, \quad p_{d+1} - \zeta_{d+1}, \quad p_{d+2} - \zeta_{d+2}, \quad \cdots, \quad p_D - \zeta_D)
\end{array}
\tag{33}
$$

Apparently, this equals to prove:

$$
\begin{aligned}
p_1 + p_1 \mathbb{E}(p_2 + \zeta_2, \quad p_3 + \zeta_3, \quad \cdots, \quad p_d + \zeta_d + \zeta_1, \quad p_{d+1} - \zeta_{d+1}, \quad p_{d+2} - \zeta_{d+2}, \quad \cdots, \quad p_D - \zeta_D) \\
\geq \\
p_1 + p_1 \mathbb{E}(p_2, \quad p_3 + \zeta_3, \quad \cdots, \quad p_d + \zeta_d + \zeta_1 + \zeta_2, \quad p_{d+1} - \zeta_{d+1}, \quad p_{d+2} - \zeta_{d+2}, \quad \cdots, \quad p_D - \zeta_D)
\end{aligned}
\tag{34}
$$

which can be degenerated to Inequality 30 by removing $p_1$ from both auxiliary sequences.

Using the same method, we can iteratively transfer each $\zeta_i$ from $p_i$ to $p_d$ for $i = 1, 2, \cdots, d$, ensuring that each step maintains the inequality. Consequently, all inequalities in the chain (29) hold, except the last.

To conclude, after transferring all $\zeta_i$ from $i = 1$ to $d$, we arrive at the following:

$$
\begin{aligned}
\mathbb{E}(p_1 + \zeta_1, \quad p_2 + \zeta_2, \quad p_3 + \zeta_3, \quad \cdots, \quad p_d + \zeta_d, \quad p_{d+1} - \zeta_{d+1}, \quad p_{d+2} - \zeta_{d+2}, \quad \cdots, \quad p_D - \zeta_D) \\
\geq \\
\mathbb{E}(p_1, \quad p_2, \quad p_3, \quad \cdots, \quad p_d + \sum_{i=1}^{d} \zeta_i, \quad p_{d+1} - \zeta_{d+1}, \quad p_{d+2} - \zeta_{d+2}, \quad \cdots, \quad p_D - \zeta_D)
\end{aligned}
\tag{35}
$$

This completes the proof for the first part of the inequality chain.

**Second Part of the Inequality Chain**

We now address the final inequality in the chain (29), specifically:

$$
\begin{aligned}
\mathbb{E}(p_1, \quad p_2, \quad p_3, \quad \cdots, \quad p_d + \sum_{i=1}^{d} \zeta_i, \quad p_{d+1} - \zeta_{d+1}, \quad p_{d+2} - \zeta_{d+2}, \quad \cdots, \quad p_D - \zeta_D) \\
\geq \\
\mathbb{E}(p_1, \quad p_2, \quad p_3, \quad \cdots, \quad p_d + \sum_{i=1}^{d} \zeta_i, \quad p_{d+1} - \sum_{i=d+1}^{D} \zeta_i, \quad p_{d+2}, \quad \cdots, \quad p_D)
\end{aligned}
\tag{36}
$$

Let $\Delta E_5$ denote the difference between the left-hand side and the right-hand side of the above inequality:

$$
\begin{aligned}
\Delta E_5 = {} & \mathbb{E}(p_1, p_2, p_3, \cdots, p_d + \sum_{i=1}^{d} \zeta_i, p_{d+1} - \zeta_{d+1}, p_{d+2} - \zeta_{d+2}, \cdots, p_D - \zeta_D) \\
& - \mathbb{E}(p_1, p_2, p_3, \cdots, p_d + \sum_{i=1}^{d} \zeta_i, p_{d+1} - \sum_{i=d+1}^{D} \zeta_i, p_{d+2}, \cdots, p_D) \\
= {} & (\prod_{i=1}^{d-1} p_i)(p_d + \sum_{i=1}^{d} \zeta_i) \left[ \sum_{k=d+1}^{D} ( \prod_{i=d+1}^{k} (p_i - \zeta_i)) - \sum_{k=d+1}^{D} ( \prod_{i=d+1}^{k} P_i') \right]
\end{aligned}
\tag{37}
$$

Note that the multiplicative coefficient $(\prod_{i=1}^{d-1} p_i)(p_d + \sum_{i=1}^{d} \zeta_i)$ is always positive, as probabilities are non-negative. Therefore, the sign of $\Delta E_5$ depends solely on the later bracketed difference. We denote this difference as $\Delta E_5'$:

$$\Delta E_5' = \sum_{k=d+1}^{D} ( \prod_{i=d+1}^{k} (p_i - \zeta_i)) - \sum_{k=d+1}^{D} ( \prod_{i=d+1}^{k} P_i')$$

$$= \sum_{k=d+1}^{D} ( \prod_{i=d+1}^{k} (p_i - \zeta_i)) - \left[ (p_{d+1} - \zeta) + \sum_{k=d+2}^{D} ( \prod_{i=d+2}^{k} p_i)(p_{d+1} - \zeta) \right]$$

$$= \left[ (p_{d+1} - \zeta_{d+1}) + \sum_{k=d+2}^{D} ( \prod_{i=d+1}^{k} (p_i - \zeta_i)) \right] - \left[ (p_{d+1} - \zeta) + \sum_{k=d+2}^{D} ( \prod_{i=d+2}^{k} p_i)(p_{d+1} - \zeta) \right]$$

$$= (\zeta - \zeta_{d+1}) + \sum_{k=d+2}^{D} \left[ \prod_{i=d+1}^{k} (p_i - \zeta_i) - ( \prod_{i=d+2}^{k} p_i)(p_{d+1} - \zeta) \right]$$

$$= (\zeta - \zeta_{d+1}) + \sum_{k=d+2}^{D} \left[ \zeta( \prod_{i=d+2}^{k} p_i) + \prod_{i=d+1}^{k} (p_i - \zeta_i) - \prod_{i=d+1}^{k} p_i \right]$$

$$= (\zeta - \zeta_{d+1}) + \sum_{k=d+2}^{D} \left[ \zeta( \prod_{i=d+2}^{k} p_i) + \sum_{i=d+1}^{k} (-\zeta_i)( \prod_{\substack{j=d+1 \\ j \neq i}}^{k} p_j) + R_2(\zeta_{d+1}, \zeta_{d+2}, \cdots, \zeta_k) \right], \tag{38}$$

where in the last two terms of Eq. (38), we split the result of $\prod_{i=d+1}^{k}(p_i - \zeta_i) - \prod_{i=d+1}^{k} p_i$ into two terms. The first term represents the summation of all elements with only one $\zeta_i$, and $R_2(\zeta_{d+1}, \zeta_{d+2}, \cdots, \zeta_k)$ denotes the sum of all possible products involving two or more distinct $\zeta_i$ terms from $\{\zeta_{d+1}, \zeta_{d+2}, \cdots, \zeta_k\}$. Now we want to prove that $R_2(\zeta_{d+1}, \zeta_{d+2}, \cdots, \zeta_k) \geq 0$. Since $R_2(\zeta_{d+1}, \zeta_{d+2}, \cdots, \zeta_k)$ only exists when $k \geq d + 2$, we define it as follow:

$$R_2(\zeta_{d+1}, \zeta_{d+2}, \cdots, \zeta_k) = \prod_{i=d+1}^{k} (p_i - \zeta_i) - \left[ \prod_{i=d+1}^{k} p_i - \sum_{i=d+1}^{k} \zeta_i \prod_{\substack{j=d+1 \\ j \neq i}}^{k} p_j \right], \quad k \geq d + 2. \tag{39}$$

Then, we can calculate the partial derivative of the function $R_2(\zeta_{d+1}, \zeta_{d+2}, \cdots, \zeta_k)$ with respect to $\zeta_m$:

$$\frac{d(R_2(\zeta_{d+1}, \zeta_{d+2}, \cdots, \zeta_k))}{d(\zeta_m)} = - \prod_{\substack{i=d+1 \\ i \neq m}}^{k} (p_i - \zeta_i) + \prod_{\substack{j=d+1 \\ j \neq m}}^{k} p_j. \tag{40}$$

Given $0 \leq p_i - \zeta_i \leq p_i \leq 1$, we observe that $\frac{d(R_2(\zeta_{d+1}, \zeta_{d+2}, \ldots, \zeta_k))}{d(\zeta_m)} \geq 0$ is always true, which implies that $R_2(\zeta_{d+1}, \zeta_{d+2}, \ldots, \zeta_k)$ is monotonically non-decreasing with respect to $\zeta_m$ for all $m \in \{d + 1, d + 2, \ldots, k\}$. Note that $R_2(\zeta_{d+1}, \zeta_{d+2}, \ldots, \zeta_k)$ is differentiable on $(0, 1)$ and continuous on $[0, 1]$ with respect to all $\zeta_m$. Therefore, $R_2(\zeta_{d+1}, \zeta_{d+2}, \ldots, \zeta_k)$ attains its minimum value when $\zeta_m = 0$ for all $m \in \{d + 1, \ldots, k\}$. Since $R_2(0, 0, \ldots, 0) = 0$, we conclude that $R_2(\zeta_{d+1}, \zeta_{d+2}, \ldots, \zeta_k) \geq 0$.

Then

$$\Delta E_5' \geq (\zeta - \zeta_{d+1}) + \sum_{k=d+2}^{D} \left[ \zeta( \prod_{i=d+2}^{k} p_i) - \sum_{i=d+1}^{k} \zeta_i( \prod_{\substack{j=d+1 \\ j \neq i}}^{k} p_j) \right]$$

$$= (\zeta - \zeta_{d+1}) + \sum_{k=d+2}^{D} \left[ ( \sum_{i=d+1}^{D} \zeta_i)( \prod_{i=d+2}^{k} p_i) - \sum_{i=d+1}^{k} \zeta_i( \prod_{\substack{j=d+1 \\ j \neq i}}^{k} p_j) \right]. \tag{41}$$

When $D - d = 2$:

$$\Delta E_5'^{[2]} = (\zeta - \zeta_{d+1}) + \sum_{k=d+2}^{d+2} \left[ (\prod_{i=d+2}^{k} p_i)\zeta - \sum_{i=d+1}^{k} \zeta_i (\prod_{\substack{j=d+1 \\ j \neq i}}^{k} p_j) \right]$$

$$= \zeta_{d+2} + p_{d+2}\zeta - \zeta_{d+1}p_{d+2} - \zeta_{d+2}p_{d+1}$$

$$= \zeta_{d+2}(1 - p_{d+1}) + (\zeta - \zeta_{d+1})p_{d+2}$$

$$\geq (\zeta - \zeta_{d+1})p_{d+2}. \tag{42}$$

Since $0 \leq p_i \leq 1$ and $0 \leq \zeta_i \leq 1$, it is clear that $\Delta E_5'^{[2]} \geq 0$.

Similarly, When $D - d = 3$:

$$\Delta E_5'^{[3]} = \Delta E_5'^{[2]} + \left[ (\prod_{i=d+2}^{d+3} p_i)\zeta - \sum_{i=d+1}^{d+3} \zeta_i (\prod_{\substack{j=d+1 \\ j \neq i}}^{d+3} p_j) \right]$$

$$\geq (\zeta - \zeta_{d+1})p_{d+2} + (p_{d+2}p_{d+3}\zeta - \zeta_{d+1}p_{d+2}p_{d+3} - \zeta_{d+2}p_{d+1}p_{d+3} - \zeta_{d+3}p_{d+1}p_{d+2})$$

$$= (\zeta_{d+2} + \zeta_{d+3})p_{d+2} + (p_{d+2}p_{d+3}\zeta - \zeta_{d+1}p_{d+2}p_{d+3} - \zeta_{d+2}p_{d+1}p_{d+3} - \zeta_{d+3}p_{d+1}p_{d+2})$$

$$= \zeta_{d+2}(p_{d+2} - p_{d+1}p_{d+3}) + \zeta_{d+3}(p_{d+2} - p_{d+1}p_{d+2}) + (\zeta - \zeta_{d+1})p_{d+2}p_{d+3}$$

$$\geq \zeta_{d+2}(p_{d+2} - p_{d+1}p_{d+2}) + \zeta_{d+3}(p_{d+2} - p_{d+1}p_{d+2}) + (\zeta - \zeta_{d+1})p_{d+2}p_{d+3}$$

$$\geq (\zeta - \zeta_{d+1})p_{d+2}p_{d+3}$$

$$\geq 0. \tag{43}$$

Using induction, assume that for any $D - d = k \in [2, \infty)$, $\Delta E_5'^{[k]} = (\zeta - \zeta_{d+1}) \prod_{i=d+2}^{d+k} p_i \geq 0$.

Then for $D - d = k + 1$:

$$\Delta E_5'^{[k+1]} \geq (\zeta - \zeta_{d+1}) \prod_{i=d+2}^{d+k} p_i + (\prod_{i=d+2}^{d+k+1} p_i\zeta - \sum_{i=d+1}^{d+k+1} \zeta_i (\prod_{\substack{j=d+1 \\ j \neq i}}^{d+k+1} p_j))$$

$$= (\zeta - \zeta_{d+1}) \prod_{i=d+2}^{d+k} p_i + \left( \zeta(\prod_{i=d+2}^{d+k+1} p_i) - \zeta_{d+1}(\prod_{j=d+2}^{d+k+1} p_j) - \zeta_{d+2}(\prod_{\substack{j=d+1 \\ j \neq d+2}}^{d+k+1} p_j) - \cdots - \zeta_{d+k+1}(\prod_{\substack{j=d+1 \\ j \neq d+k+1}}^{d+k+1} p_j) \right)$$

$$= (\zeta_{d+2} + \zeta_{d+3} + \cdots + \zeta_{d+k+1}) \prod_{i=d+2}^{d+k} p_i +$$

$$\left( \zeta(\prod_{i=d+2}^{d+k+1} p_i) - \zeta_{d+1}(\prod_{j=d+2}^{d+k+1} p_j) - \zeta_{d+2}(\prod_{\substack{j=d+1 \\ j \neq d+2}}^{d+k+1} p_j) - \cdots - \zeta_{d+k+1}(\prod_{\substack{j=d+1 \\ j \neq d+k+1}}^{d+k+1} p_j) \right)$$

$$= \zeta_{d+2} \left( \prod_{i=d+2}^{d+k} p_i - \prod_{\substack{j=d+1 \\ j \neq d+2}}^{d+k+1} p_j \right) + \zeta_{d+3} \left( \prod_{i=d+2}^{d+k} p_i - \prod_{\substack{j=d+1 \\ j \neq d+3}}^{d+k+1} p_j \right) + \cdots + \zeta_{d+k+1} \left( \prod_{i=d+2}^{d+k} p_i - \prod_{\substack{j=d+1 \\ j \neq d+k+1}}^{d+k+1} p_j \right)$$

$$+ (\zeta - \zeta_{d+1}) \prod_{i=d+2}^{d+k+1} p_i. \tag{44}$$

For the first term in Eq. (44), we observe:

$$
\begin{aligned}
\left( \prod_{i=d+2}^{d+k} p_i - \prod_{\substack{j=d+1 \\ j \neq d+2}}^{d+k+1} p_j \right) &= \left( \prod_{i=d+2}^{d+k} p_i - \prod_{\substack{j=d+1 \\ j \neq d+2}}^{d+k} p_j \cdot p_{d+k+1} \right) \\
&\geq \left( \prod_{i=d+2}^{d+k} p_i - \prod_{\substack{j=d+1 \\ j \neq d+2}}^{d+k} p_j \cdot p_{d+2} \right) \\
&= \left( \prod_{i=d+2}^{d+k} p_i - \prod_{j=d+1}^{d+k} p_j \right) \\
&= (1 - p_{d+1}) \left( \prod_{i=d+2}^{d+k} p_i \right) \geq 0.
\end{aligned}
\tag{45}
$$

By the same reasoning, we can show that all coefficients in Eq. (44) are non-negative, and we have:

$$
\begin{aligned}
\Delta E_5'^{[k+1]} &\geq (\zeta - \zeta_{d+1}) \prod_{i=d+2}^{d+k+1} p_i \\
&\geq 0.
\end{aligned}
\tag{46}
$$

Therefore, we can conclude that $\Delta E_5' \geq 0$, and thus:

$$
\Delta E_5 \geq 0.
\tag{47}
$$

**Conclusion:**

Since:

$$
\Delta E_4 \geq 0, \quad \Delta E_5 \geq 0,
\tag{48}
$$

we conclude that the inequality chain (29) holds, which means:

$$
\mathbb{E}[L]_{\text{improved}} \geq \mathbb{E}[L]_{\text{concentrate}}.
\tag{49}
$$

### A.3. Proving that $\mathbb{E}[L]_{\textbf{improved}} \geq \mathbb{E}[L]_{\textbf{original}}$

Given that:

$$
\mathbb{E}[L]_{\text{improved}} \geq \mathbb{E}[L]_{\text{concentrate}}, \quad \mathbb{E}[L]_{\text{concentrate}} \geq \mathbb{E}[L]_{\text{original}},
\tag{50}
$$

we establish that:

$$
\mathbb{E}[L]_{\text{improved}} \geq \mathbb{E}[L]_{\text{original}}.
\tag{51}
$$

### A.4. Cases of $p_d + \zeta > 1$

At the beginning of the proof, we assume $p_d + \zeta \leq 1$. In cases where $p_d + \zeta > 1$, the parameter $\zeta$ can be distributed across multiple positions to satisfy the constraints. Specifically, we divide $\zeta$ into $n$ parts:

$$\zeta = \hat{\zeta}_d + \hat{\zeta}_{d-1} + \hat{\zeta}_{d-2} + \cdots + \hat{\zeta}_{d-n+1}, \tag{52}$$

where

$$p_d + \hat{\zeta}_d = 1, \tag{53}$$

$$p_{d-1} + \hat{\zeta}_{d-1} = 1, \tag{54}$$

$$\cdots, \tag{55}$$

$$p_{d-n+2} + \hat{\zeta}_{d-n+2} = 1, \tag{56}$$

$$p_{d-n+1} + \hat{\zeta}_{d-n+1} < 1. \tag{57}$$

These adjustments are applied to $n$ positions as follows:

$$P_i' = \begin{cases} p_i + \hat{\zeta}_i, & i = d-n+1, \cdots, d \\ p_i - \zeta, & i = d+1 \\ p_i, & otherwise, \end{cases} \qquad s.t. \ \ \zeta = \sum_{i=1}^{d} \zeta_i = \sum_{i=d+1}^{D} \zeta_i = \sum_{i=d-n+1}^{d} \hat{\zeta}_i. \tag{58}$$

Next, we construct the following inequality chain:

$$
\begin{aligned}
&\mathbb{E}(p_1, \quad p_2, \quad p_3, \quad \cdots, \quad p_d, \quad p_{d+1}, \quad p_{d+2}, \quad \cdots, \quad p_D) \\
&\leq \\
&\mathbb{E}(p_1, \quad p_2, \quad p_3, \quad \cdots, \quad p_d + \hat{\zeta}_d, \quad p_{d+1} - \hat{\zeta}_d, \quad p_{d+2}, \quad \cdots, \quad p_D) \\
&\leq \\
&\mathbb{E}(p_1, \quad p_2, \quad \cdots, \quad p_{d-1} + \hat{\zeta}_{d-1}, \quad p_d + \hat{\zeta}_d, \quad p_{d+1} - \hat{\zeta}_d - \hat{\zeta}_{d-1}, \quad p_{d+2}, \quad \cdots, \quad p_D) \\
&\leq \\
&\cdots \\
&\leq \\
&\mathbb{E}(p_1, \cdots, \quad p_{d-n+1} + \hat{\zeta}_{d-n+1}, \quad \cdots, \quad p_{d-1} + \hat{\zeta}_{d-1}, \quad p_d + \hat{\zeta}_d, \quad p_{d+1} - \sum_{i=d-n+1}^{d} \hat{\zeta}_i, \quad p_{d+2}, \cdots, p_D)
\end{aligned}
\tag{59}
$$

Each step in the inequality chain (59) can be proven using the same method as proving $\mathbb{E}[L]_{\text{concentrate}} \geq \mathbb{E}[L]_{\text{original}}$.
Next, we construct another inequality:

$$
\begin{aligned}
&\mathbb{E}(p_1, \cdots, \quad p_{d-n+1} + \hat{\zeta}_{d-n+1}, \quad \cdots, \quad p_{d-1} + \hat{\zeta}_{d-1}, \quad p_d + \hat{\zeta}_d, \quad p_{d+1} - \sum_{i=d-n+1}^{d} \hat{\zeta}_i, \quad p_{d+2}, \cdots, p_D) \\
&\leq \\
&\mathbb{E}(p_1, \cdots, \quad p_{d-n+1} + \hat{\zeta}_{d-n+1}, \quad \cdots, \quad p_{d-1} + \hat{\zeta}_{d-1}, \quad p_d + \hat{\zeta}_d, \quad p_{d+1} - \zeta_{d+1}, \quad p_{d+2} - \zeta_{d+2}, \cdots, p_D - \zeta_D)
\end{aligned}
\tag{60}
$$

The inequality (60) can be proven using the same method as proving inequality (36).

Finally, we construct a third inequality:

$$
\mathbb{E}(p_1, \cdots, \quad p_{d-n+1} + \hat{\zeta}_{d-n+1}, \quad \cdots, \quad p_{d-1} + \hat{\zeta}_{d-1}, \quad p_d + \hat{\zeta}_d, \quad p_{d+1} - \zeta_{d+1}, \quad p_{d+2} - \zeta_{d+2}, \cdots, p_D - \zeta_D)
$$
$$
\leq \tag{61}
$$
$$
\mathbb{E}(p_1 + \zeta_1, \quad p_2 + \zeta_2, \quad \cdots, \quad p_d + \zeta_d, \quad p_{d+1} - \zeta_{d+1}, \quad p_{d+2} - \zeta_{d+2}, \cdots, p_D - \zeta_D)
$$

To prove inequality (61), we can follow the same method as proving the first to the second-to-last line in inequality (29). In that proof, we observed that each step involved moving a non-negative value $\zeta_i$ from an earlier position to a later position, and the validity of the inequality was independent of the specific value of $\zeta_i$ or the positions involved. In the case of inequality (61), we have $\zeta_1, \cdots, \zeta_d \geq 0$ and $\zeta_{d-n+1}, \cdots, \zeta_d \leq \hat{\zeta}_{d-n+1}, \cdots, \hat{\zeta}_d$. This means that we are essentially moving positive values from earlier positions to later positions, similar to the process in inequality (29). Therefore, we can use the same method to establish inequality (61).

With inequalities (61), (60) and the inequality chain (59) established, we conclude that $\mathbb{E}[L]_{\text{improved}} \geq \mathbb{E}[L]_{\text{original}}$. Therefore, we finish the proof of Theorem 3.1 in the main paper.

## B. Experiment Results on A100

This section gives the inference results on a single NVIDIA A100 GPU. It is reasonable that there are different final speedups on MI250 and A100. However, note that the mean average tokens ($\tau$) are also slightly different. This is because of the use of FP16 precision during inference, and different FP16 processing mechanisms in MI250 and A100 GPUs. Using FP32 precision yields identical $\tau$ values across both platforms, yet significantly increases the inference latency.

*Table 4.* Speedup ratios and mean accepted tokens ($\tau$) of different methods on NVIDIA A100. V represents Vicuna, L2 represents LLaMA2-Chat, and L3 represents LLaMA3-Instruct. We present the results of different methods across six datasets. *Mean* represents the average performance across these six datasets.

| Model | Method | MT-Bench Speedup | $\tau$ | HumanEval Speedup | $\tau$ | GSM8K Speedup | $\tau$ | Alpaca Speedup | $\tau$ | CNN/DM Speedup | $\tau$ | Natural Ques. Speedup | $\tau$ | Mean Speedup | $\tau$ |
|---|---|---|---|---|---|---|---|---|---|---|---|---|---|---|---|
| | | | | | | Temperature=0 | | | | | | | | | |
| V 7B | Eagle-2 | 3.30× | 5.03 | 3.59× | 5.36 | 3.21× | 4.94 | 3.00× | **4.86** | 2.57× | 4.10 | 2.48× | **3.82** | 3.02× | 4.69 |
| | Gumiho(ours) | **3.62×** | **5.22** | **4.22×** | **5.81** | **3.48×** | **5.07** | **3.13×** | 4.82 | **2.91×** | **4.46** | **2.60×** | 3.80 | **3.33×** | **4.86** |
| V 13B | Eagle-2 | 3.05× | 4.92 | 3.60× | 5.42 | 3.24× | 4.79 | 2.99× | 4.90 | 2.47× | 4.21 | 2.48× | 3.71 | 2.98× | 4.66 |
| | Gumiho(ours) | **3.33×** | **5.23** | **4.12×** | **6.01** | **3.35×** | **5.08** | **3.07×** | **4.97** | **2.68×** | **4.40** | **2.52×** | **3.81** | **3.19×** | **4.92** |
| L2 7B | Eagle-2 | 3.28× | 4.76 | 3.71× | 5.38 | 3.28× | 4.77 | 3.17× | **4.66** | 2.61× | 4.09 | 2.78× | 4.16 | 3.14× | 4.64 |
| | Gumiho(ours) | **3.44×** | **4.92** | **3.92×** | **5.57** | **3.42×** | **4.83** | **3.19×** | 4.54 | **2.82×** | **4.20** | **2.91×** | **4.19** | **3.28×** | **4.71** |
| L2 13B | Eagle-2 | 3.02× | 4.77 | 3.60× | 5.53 | 3.40× | 4.89 | 3.10× | 4.60 | 2.54× | 4.26 | 2.77× | 4.12 | 3.07× | 4.69 |
| | Gumiho(ours) | **3.24×** | **4.97** | **3.78×** | **5.85** | **3.61×** | **5.03** | **3.19×** | **4.63** | **2.72×** | **4.38** | **2.85×** | **4.20** | **3.23×** | **4.84** |
| L3 8B | Eagle-2 | 2.71× | 4.35 | 3.20× | 5.06 | 2.79× | 4.47 | 2.78× | 4.87 | 2.24× | 3.81 | 2.26× | 3.53 | 2.67× | 4.35 |
| | Gumiho(ours) | **2.95×** | **4.48** | **3.62×** | **5.22** | **3.20×** | **4.62** | **2.86×** | **4.88** | **2.39×** | **3.90** | **2.41×** | **3.62** | **2.91×** | **4.45** |
| | | | | | | Temperature=1 | | | | | | | | | |
| V 7B | Eagle-2 | 2.73× | 4.32 | 2.99× | 4.65 | 2.59× | 4.41 | 2.55× | **4.25** | 2.28× | 3.87 | 2.21× | 3.52 | 2.56× | 4.17 |
| | Gumiho(ours) | **3.00×** | **4.35** | **3.30×** | **4.82** | **2.98×** | **4.54** | **2.61×** | 4.22 | **2.48×** | **3.94** | **2.29×** | **3.63** | **2.78×** | **4.25** |
| V 13B | Eagle-2 | 2.73× | 4.35 | 3.14× | 4.85 | 2.81× | 4.54 | 2.75× | 4.57 | 2.34× | 4.01 | 2.26× | 3.53 | 2.68× | 4.31 |
| | Gumiho(ours) | **2.92×** | **4.56** | **3.46×** | **5.31** | **2.95×** | **4.63** | **2.89×** | **4.67** | **2.42×** | **4.05** | **2.43×** | **3.70** | **2.85×** | **4.49** |
| L2 7B | Eagle-2 | 2.94× | 4.53 | 3.32× | 5.10 | 3.01× | 4.69 | 2.80× | **4.61** | 2.50× | 3.92 | 2.60× | 4.04 | 2.86× | 4.48 |
| | Gumiho(ours) | **3.02×** | **4.65** | **3.41×** | **5.31** | **3.12×** | **4.74** | **2.97×** | 4.43 | **2.65×** | **4.07** | **2.83×** | **4.12** | **3.00×** | **4.55** |
| L2 13B | Eagle-2 | 2.80× | 4.61 | 3.41× | 5.37 | 3.19× | 4.75 | 2.93× | **4.50** | 2.41× | 4.15 | 2.62× | 4.06 | 2.90× | 4.57 |
| | Gumiho(ours) | **2.96×** | **4.85** | **3.71×** | **5.72** | **3.33×** | **4.88** | **2.99×** | **4.50** | **2.54×** | **4.31** | **2.71×** | **4.12** | **3.04×** | **4.73** |
| L3 8B | Eagle-2 | 2.34× | 3.92 | 2.82× | 4.81 | 2.61× | 4.36 | 2.53× | 4.50 | 2.04× | 3.55 | 2.02× | 3.36 | 2.39× | 4.08 |
| | Gumiho(ours) | **2.52×** | **4.11** | **3.13×** | **4.97** | **2.81×** | **4.62** | **2.67×** | **4.54** | **2.19×** | **3.66** | **2.12×** | **3.47** | **2.57×** | **4.23** |

## C. Training Details and Hyper-parameters

Similar to Eagle-2 (Li et al., 2024b), we employ both the regression loss $\mathcal{L}_{reg}$ and the classification loss $\mathcal{L}_{cls}$ to train the draft model. The total loss $\mathcal{L}$ is defined as a weighted combination of these two components:

$$\mathcal{L} = w_{reg} \cdot \mathcal{L}_{reg} + w_{cls} \cdot \mathcal{L}_{cls},$$

where $w_{reg}$ and $w_{cls}$ denote the weighting coefficients for the regression loss $\mathcal{L}_{reg}$ and the classification loss $\mathcal{L}_{cls}$, respectively.

Similar to EAGLE-2's tree attention, we select the top 10 output tokens from each Transformer head as input for the subsequent head, *i.e.*, topk=10. For FTA, we extract the top 35 output tokens from each MLP head , *i.e.*, s=35. Hyper-parameters can be found in Tab. 5.

*Table 5.* Hyper-parameter configurations of Gumiho.

| Hyper-parameters | Vicuna 7B/13B LLaMA2 7B/13B LLaMA3 8B | LLaMA2 70B LLaMA3 70B |
|---|---|---|
| Learning rate | 2e-4 | 1e-4 |
| Transformer layer number | 2 | |
| MLP head number | 5 | |
| Batch size | 4 | |
| $w_{cls}$ | 0.1 | |
| $w_{reg}$ | 1 | |
| Training epoch | 10 | |
| Optimizer | AdamW | |
| $(\beta_1, \beta_2)$ | (0.9, 0.95) | |
| Per MLP structure | [2*hidden_state, 1*hidden_state]*1 $\rightarrow$ [1*hidden_state, 1*hidden_state]*5 | |
| topk | 10 | |
| s | 35 | |

## D. Ablation Study on Head Accuracy

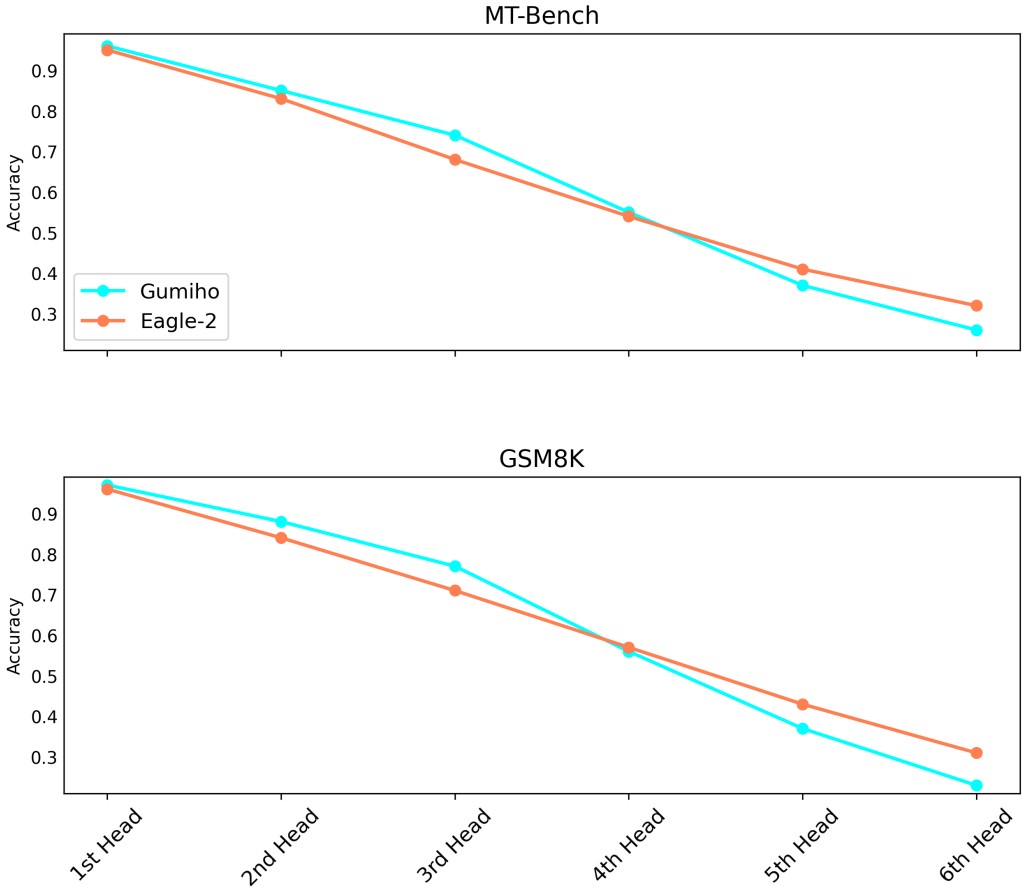

*Figure 6.* Comparison of draft head accuracy on two datasets (MT-Bench and GSM8K). Both results are based on Vicuna 7B with the temperature set to 0.

We conducted a comparative analysis of head-wise accuracy between our method and EAGLE-2 based on Vicuna 7B with temperature set to 0 on MT-Bench and GSM8K datasets. It should be noted that we have a total of seven draft heads, while EAGLE-2 only has six heads. Therefore, to facilitate comparison, we have only conducted an accuracy comparison between the first six heads of ours and the six heads of EAGLE-2. As illustrated in Fig. 6, our approach enhances the accuracy of front heads, which are responsible for generating the initial tokens in the draft sequence. The precision of these early tokens substantially impacts the final mean accepted tokens($\tau$). Our back heads employ a parallel MLP architecture, resulting in lower accuracy compared to EAGLE-2. This accuracy distribution aligns with our theoretical findings. Our theorem demonstrates that optimizing the accuracy distribution across heads, specifically through enhancing precision in front heads while proportionally reducing accuracy in back heads, leads to better overall mean accepted tokens($\tau$).

