# OpenReview forum: "Gumiho: A Hybrid Architecture to Prioritize Early Tokens in Speculative Decoding"
_ICML.cc/2025/Conference — ICML 2025 poster_

### Official Review · Reviewer_1s98 · 2025-02-28

**Overall Recommendation:** 4

**Summary:**

Previous speculative decoding methods treat all tokens in a sequence equally, while this paper demonstrates that earlier tokens are more critical for success. To improve efficiency, the authors propose Gumiho, a hybrid model that prioritizes early tokens with a serial two-layer Transformer and uses lightweight parallel MLP heads for later tokens. This strategy boosts both the accuracy of early token predictions and the efficiency of later ones. Experimental results show that Gumiho outperforms existing methods in terms of speedup ratio and acceptance length.

**Claims And Evidence:**

The claims made in the submission are supported by clear and convincing evidence.

**Essential References Not Discussed:**

No.

**Experimental Designs Or Analyses:**

I checked the soundness/validity of experimental designs and analyses, which is section 4 in the paper.

**Methods And Evaluation Criteria:**

Proposed methods and evaluation criteria make sense for the problem or application at hand.

**Other Comments Or Suggestions:**

See weaknesses.

**Other Strengths And Weaknesses:**

Pros:

1. The finding of the proof is solid and inspiring.

2. Given the proof, the approach in this paper is reasonable.

3. The speed-up ratio gains, especially on large-scale LLMs such as LLaMA2 70B and LLaMA3 70B are obvious compared to Eagle2, which shows the effectiveness of the proposed method.

Cons:

1. There is a lack of comparison to the latest state-of-the-art papers, which propose better results than Eagle2, such as HASS [1] and DDD [2].

2. The method uses more parameters in the draft model compared to Eagle, Eagle2, and Medusa. Does the training process consume more GPU memory?

[1] Learning Harmonized Representations for Speculative Sampling, ICLR, 2025.

[2] Dynamic Depth Decoding: Faster Speculative Decoding for LLMs, arXiv, 2024.

**Questions For Authors:**

1. Is it able to combine the proposed method with the state-of-the-art Spd method to further improve the speed up ratio? It is better if the conclusion of the theorem can be applied to other methods.

2. What is the average draft time for each part of the method? In figure3 you show the overall draft time including the model forward time and full tree attention time. I would like to see each part of them.

**Relation To Broader Scientific Literature:**

The proposed Gumiho combines the serial heads in Eagle2 and parallel heads in Medusa.

**Theoretical Claims:**

I checked the correctness of the proofs for theoretical claims, which is theorem 3.1 and appendix A in the paper.

---

> ### Author Rebuttal · Authors · 2025-03-31
>
> Thank you for your suggestions and support.
>
> **Q1. There is a lack of comparison to the latest state-of-the-art papers, which propose better results than Eagle2, such as HASS [1] and DDD [2]**
>
> A1:**(1) Hass** improves Eagle2 by addressing the training-inference inconsistency in serial transformer heads within SPD. While Hass focuses on training methodology, our work enhances performance via architectural modifications. These two directions are orthogonal: our method could integrate Hass’s training improvements for further gains.
>
> In the table below, we compare the performance of our Gumiho with Hass. Additionally, we evaluate a variant of our model (denoted as Gumiho w/ Hass ) where Hass’s training method is applied to our Transformer Head. All results are conducted on a Mi250 GPU using the LLaMA3-8B model with a temperature of 0.
>
> || MT-Bench|HumanEval|
> |-|-|-|
> |Eagle2|2.16×|2.51×|
> |Hass|2.26×|2.59×|
> |Gumiho|2.38×|2.77×|
> |Gumiho w/ Hass|**2.43×**|**2.84×**|
>
> **(2) DDD** improves Eagle2 by using a dynamic draft tree. Since DDD’s code is not open-sourced, we compare relative improvements over Eagle2. On the MT-Bench dataset, the performance gains of DDD and Gumiho are:
>
> ||DDD|Gumiho|
> |-|-|-|
> |V 7B|3.5%|**9.4%**|
> |V 13B|4.2%|**6.3%**|
> |L2 7B|2.3%|**5.5%**|
> |L2 13B|2.3%|**5.4%**|
>
> The aforementioned results and discussions will be included in the final version of our paper.
>
> **Q2. The method uses more parameters in the draft model compared to Eagle, Eagle2, and Medusa. Does the training process consume more GPU memory?**
>
> A2: Yes, you are correct. Our method does require more GPU memory during training compared to Eagle, Eagle2, and Medusa. Here’s why:
> - Eagle and Eagle2 use one single-layer Transformer head and reuse it serially across multiple speculative steps.
> - Medusa only maintains five parallel MLP heads.
> - Our approach, however, employs a two-layer Transformer head alongside five parallel MLP heads, all trained simultaneously.
>
> This architectural choice increases memory consumption. However, it enables a hybrid generation strategy: serial decoding for the first few tokens (to ensure accuracy) and parallel decoding for subsequent tokens (to maximize speed). While this introduces a memory-accuracy trade-off, it achieves a significant speedup improvement. We will explicitly discuss this trade-off in the final manuscript.
>
> **Q3. Is it able to combine the proposed method with the state-of-the-art Spd method to further improve the speed up ratio? It is better if the conclusion of the theorem can be applied to other methods.**
>
> A3: As illustrated in A1, our method could integrate Hass’s training strategy for further gains.
>
> In addition to Hass, our core theoretical principle—**prioritizing the initial tokens**—can also be generalized to serial multi-head draft models (e.g., Hydra or Eagle). By reallocating Head parameters to emphasize early tokens, this principle could further enhance the speedup ratio of existing SOTA methods.
>
> Moreover, existing methods typically use a single smaller model (e.g., LLaMA-3-3B) to accelerate larger counterparts like LLaMA-3-405B. However, our analysis reveals that adopting a differentiated draft model strategy—where a more capable model (e.g., LLaMA-3-8B) predict initial tokens, while a faster but lighter model (e.g., LLaMA-3-1B) handles subsequent tokens—could optimize the quality-speed trade-off beyond current homogeneous settings.
>
> Exploring these extensions is a key direction for our future research.
>
> **Q4. What is the average draft time for each part of the method? In figure3 you show the overall draft time including the model forward time and full tree attention time. I would like to see each part of them.**
>
> A4:  The following table presents the draft time of the Vicuna 7B on Mi250 GPU.
> - `Serial Head Forward` represents `the time for a single serial head forward pass once`$\times$`the number of forward passes`. In Eagle2, a single serial head corresponds to a single-layer Transformer, while in Gumiho it represents a two-layer Transformer.
> - `Parallel Head Forward` denotes the time for a parallel head forward pass once. Eagle2 does not have parallel heads, whereas in Gumiho, this refers to the parallel MLP heads.
> - `Tree Attention / Full Tree Attention` represents the time required to construct the tree structure in Eagle2 and Gumiho.
> - `Additional Computations` represents the time for additional computing. (e.g., using torch.top_k to retrieve current tokens).
> - `Total Time` means the total time consumed for a draft generation.
>
> ||Eagle2|Gumiho|
> |-|-|-|
> |Serial Head Forward|9.8ms$\times$6|21ms$\times$2|
> |Parallel Head Forward|N/A|14.7ms|
> |Tree Attention / Full Tree Attention|3.3ms|3.4ms|
> |Additional Computations|3.1ms|3.2ms|
> |Total Time|65.2ms|63.3ms|

---

### Official Review · Reviewer_w5FY · 2025-03-04

**Overall Recommendation:** 4

**Summary:**

This paper proposes a new speculative decoding method called Gumiho. It combines the parallel draft head architecture and sequential draft head architecture to derive a hybrid architecture. The idea behind the paper is to prioritize the accuracy of the early tokens and make a rigorous mathematical proof to show that this can enhance the overall performance. The experiments on six datasets with different baseline models and temperatures show that Gumiho can achieve new state-of-the-art results.

**Claims And Evidence:**

Theorem 3.1 in the paper as well as the proof in the appendix can support the claim.

**Essential References Not Discussed:**

No

**Experimental Designs Or Analyses:**

I checked the experimental designs and analyses. The datasets and baseline models are commonly used in other papers. The results are reasonable.

**Methods And Evaluation Criteria:**

I think the core idea of prioritizing the accuracy of the early tokens under a limited computational budget is important and meaningful to the research area of SpD, especially since a rigorous theoretical proof is provided. The evaluation criteria are the commonly used datasets and baseline models.

**Other Comments Or Suggestions:**

No

**Other Strengths And Weaknesses:**

Strengths:

The paper is well-organized, making it easy to follow.

The theoretical analysis is rich and the proof is rigorous.

The experimental section is comprehensive, covering multiple architectures and settings, demonstrates the validity and robustness of the proposed method. The overall speedup improvements over Eagle is non-negligible.



Weaknesses:

The ablation study shows that using FTA has marginal improvement.

The conclusion of the theorem is simple but it seems that the proof is rather complicated. Is there an easier way to give the proof?

The theorem assumes that the total amount of decrease to the second part is the same as that of increase to the first part of the sequence. However, this is hardly the case in reality. Is it able to give a more general assumption?

**Questions For Authors:**

I wonder if the idea of ‘the prior tokens in speculative decoding are more important’ has been investigated by other papers since the conclusion is obvious and easy to derive. If so, the novelty of this paper will be decreased, and the author should give a further comparison to other papers.

**Relation To Broader Scientific Literature:**

There may have some findings on the conclusion ‘the prior tokens in speculative decoding are more important’ already.

**Theoretical Claims:**

I roughly checked the proof of Theorem 3.1, and no problem was found.

---

> ### Author Rebuttal · Authors · 2025-03-31
>
> Thank you for your suggestions and support.
>
> **Q1. The ablation study shows that using FTA has marginal improvement.**
>
> A1: The improvement contributed by our proposed FTA method is non-negligible in the context of the overall performance gain. In the FTA ablation study, the speedup ratio increases by 0.05, while the total improvement of our method over Eagle2 is 0.27. This means FTA accounts for approximately 18.5% of the total performance gain, thereby demonstrating its critical role in our approach.
>
> Moreover, FTA introduces a lossless improvement compared to existing TA  without incurring additional computational overhead. It only requires modifying the Tree Mask during verification by setting specific positions from 0 to 1 (i.e., unmasking certain tokens). This design ensures efficiency while enhancing performance.
>
> **Q2. The conclusion of the theorem is simple but it seems that the proof is rather complicated. Is there an easier way to give the proof?**
>
> A2: Note that although the conclusion of the theorem appears simple, the situation in reality is rather complicated since we have only imposed very few restrictions on the Theorem. Specifically, we only require that the sum of the increments in the former part and the decrements in the latter part must be equal. The probabilities $\lbrace p_i\rbrace_{i=1}^D$ in the original setting, the relative magnitude of increases and decreases $\zeta_i$, the length of the original sequence $D$, and the divided location $d$ of the former part and latter part are all unbounded and can change freely.
>
> Therefore, although the theorem itself appears simple, the high degree of freedom results in highly complex scenarios requiring exhaustive case analysis. Under the current assumptions adopted in our paper, each step in the proof is indeed necessary.
>
>
> **Q3. The theorem assumes that the total amount of decrease to the second part is the same as that of increase to the first part of the sequence. However, this is hardly the case in reality. Is it able to give a more general assumption?**
>
> A3: Actually, the assumption that *"the increment in the former part equals the decrement in the latter part"* can be interpreted as an upper-bound condition. In fact, our conclusion holds not only when these quantities are equal, but also when the increment in the former part exceeds the decrement in the latter part, in which the Eq.6 can be modified from
>
> $\sum_{i=1}^d \zeta_i=\sum_{j=d+1}^D \zeta_j$
>
> to
>
> $\sum_{i=1}^d \zeta_i\geq\sum_{j=d+1}^D \zeta_j$.
>
> This is very easy to prove that Theorem 3.1 still holds in this situation if we remove the excess portion in the former part without decreasing the mean accepted tokens.
>
> As for the scenario where the increment is less than the decrement, we believe that this is a more complex situation and we will investigate it in our future research.
>
> The aforementioned analysis will be added in the final version of our paper.
>
>
> **Q4. I wonder if the idea of ‘the prior tokens in speculative decoding are more important’ has been investigated by other papers since the conclusion is obvious and easy to derive. If so, the novelty of this paper will be decreased, and the author should give a further comparison to other papers.**
>
> A4: To the best of our knowledge, no prior work has proposed the idea presented in our paper. We are the first to introduce the idea that *"prior tokens are more important"* and provide rigorous theoretical proof for this concept. Furthermore, we leverage this insight to design a novel model architecture, Gumiho.
>
> Moreover, applying the principle of 'prior tokens are more important' to achieve speedup improvements is not straightforward. Simply enhancing prior token accuracy may increase computational overhead at the same time, which would prolong draft time and consequently undermine the overall speedup. In fact, we have no idea how this would affect the final speed-up ratio if we merely increase the computational budget on the prior tokens, since the mean accepted tokens and the draft time will increase together and they are two factors that play opposite roles.
>
> In contrast, our Gumiho tries our best to fix the overall computational budget (by increasing the computational budgets of prior tokens and decreasing the latter) and shows that we can improve the mean accepted tokens. We achieve this by simultaneously boosting the accuracy of prior tokens while adopting simplified head structures (i.e., MLP) for subsequent tokens with parallel generation. This architectural innovation effectively balances accuracy and efficiency, ultimately achieving superior speedup.

---

> > ### Comment · Reviewer_w5FY · 2025-04-02
> >
> > Thank you for the rebuttal.
> >
> > Although it is still unknown whether the conclusion in this paper still holds in the situation that the amount of decrease to the second part is greater than the increase of the first part, it is acceptable since it is hard to require a decrease of the total accuracy while at the same time increase the mean accepted tokens.
> >
> > Besides that, all of my concerns are well-addressed, especially my novelty concern of this paper and the improvement of FTA.
> >
> > I would like to increase my score from 3 to 4.

---

> > > ### Author Response · Authors · 2025-04-03
> > >
> > > Thank you for your feedback.
> > >
> > > We sincerely appreciate your support. We will analyze the influence of this specific scenario in our future research.

---

### Official Review · Reviewer_E4Fx · 2025-03-11

**Overall Recommendation:** 4

**Summary:**

This paper improves the self-speculative decoding methods by combining the architecture of Eagle and Medusa. The paper uses sophisticated Transformer architecture for the early draft heads in a serial configuration to improve accuracy, and multiple lightweight MLP heads operating in parallel to enhance efficiency. The reason behind this combination is that the prior tokens in speculative decoding are more important. The paper gives a theoretical analysis of this idea. The experimental results demonstrate the effectiveness of this paper.

## update after rebuttal
I will keep my ratings since the rebuttal solve most of my concerns.

**Claims And Evidence:**

The proposed method is evaluated on multiple architectures and settings, including experiments on Vicuna, LLaMA2 and LLaMA3. It is also compared against other similar competitor methods. The results demonstrate the effectiveness and efficiency of the proposed approach.

**Essential References Not Discussed:**

There are some discussions of the accuracies of different heads in paper [1], which shows that previous heads have higher accuracy, and I think the author should have a discussion in their paper.

[1] Cerberus: Efficient Inference with Adaptive Parallel Decoding and Sequential Knowledge Enhancement.

**Experimental Designs Or Analyses:**

I have reviewed the experimental design in the paper, including experiments on multiple architectures and settings and the ablation study. In the ablation study, the authors explore FTA but show marginal improvement.

**Methods And Evaluation Criteria:**

The proposed method and evaluation criteria are reasonable and follow widely accepted evaluation standards. The benchmark datasets used are also standard.

**Other Comments Or Suggestions:**

See weaknesses above.

**Other Strengths And Weaknesses:**

Strengths:

1. I believe that this paper is meaningful to the research area of speculative decoding. The paper rigorously proves that improving the performance of the previous draft models with the expense of the later models is beneficial to the overall mean-accept-tokens of SpD.

2. The paper gives a nice hybrid architecture solution that uses sequential models to improve the previous tokens’ performance and uses parallel models to accelerate the later ones.

3. The paper is theoretically sound and easy to understand. The proof of the theorem is roughly correct.

4. The experimental results show that this paper can achieve a new state-of-the-art speedup compared to the previous methods.

Weaknesses:

1. It is not very clear why full tree attention works better than the tree attention methods in Eagle2, especially why this method cannot be applied in Eagle2. The authors should give an example to illustrate this better.

2. In lines 320 to 324. The authors claim that they use 1 MI250 GPU for evaluation except the 70B variant which requires 4 GPUs. However, you only use 1 NVIDIA A100 GPU for all models. The experimental setup on A100 should be further claimed.

3. There are some discussions of the accuracies of different heads in paper [1] which shows that previous heads have higher accuracy, and I think the author should have a discussion in their paper. There are several papers that proposed better results than Eagle2, such as [2], [3]. The author should compare and discuss them.

4. In the proof of theorem 3.1, the authors use an auxiliary probability sequence $P_i’$, and assume $p_d+\zeta \leq 1$ and discuss $p_d+\zeta>1$ at the end. I am confused why $p_i-\zeta < 0$ is not assumed and discussed accordingly.

5. In appendix D, the authors show the comparison of Eagle2 and Gumiho on two datasets. What about the other four datasets?

[1] Cerberus: Efficient Inference with Adaptive Parallel Decoding and Sequential Knowledge Enhancement.

[2] Dynamic Depth Decoding: Faster Speculative Decoding for LLMs.

[3] Learning Harmonized Representations for Speculative Sampling.

**Questions For Authors:**

My major concerns are weaknesses 1, 3 and 4. The answers would likely change my evaluation of the paper.

**Relation To Broader Scientific Literature:**

The conclusion that tokens are more important in the previous part of the sequence in SpD is not surprising.

However, this paper is still meaningful to society since it discusses the situation of SpD under a limited computational budget. Based on rigorously proven theory, it gives a useful solution to this situation.

**Theoretical Claims:**

I have reviewed the mathematical formulas in the paper and did not identify any issues.

---

> ### Author Rebuttal · Authors · 2025-03-31
>
> Thank you for your suggestions and support.
>
> **Q1. It is not very clear why full tree attention works better than the tree attention methods in Eagle2, especially why this method cannot be applied in Eagle2. The authors should give an example to illustrate this better.**
>
> A1:
>
> **(1) Why FTA cannot be applied to Eagle2:**
>
> Let us refer to Fig. 2 for illustration. In the upper half of Fig. 2, the tokens [shines, glows, radiates] are outputs of Head 3, while [warmly, brightly, intensely] are outputs of Head 4. In Eagle2, the heads operate serially—both Head 3 and Head 4 are single-layer Transformers, and the output of Head 4 depends on the output of Head 3. In our method, however, these heads operate in parallel (i.e., Head 3 and Head 4 correspond to MLP1 and MLP2 in Fig. 1). Since the outputs of MLP2 are independent of MLP1, the outputs of MLP1 and MLP2 can be freely combined.
> In the lower half of Fig. 2, FTA leverages the independence between outputs of later heads to fill shorter paths in the tree with longer ones. In contrast, Eagle2’s Tree Attention (TA) requires strict dependency between tokens along each path (later tokens depend on earlier ones), making cross-path filling impossible.
>
> **(2) Why FTA outperforms TA:**
>
> The final speedup improvement is achieved by either increasing the mean accepted tokens or reducing draft time. Compared to TA, FTA increases the mean accepted tokens by extending candidate path lengths without incurring additional computational overhead, thereby maintaining the same draft time as TA. This allows FTA to achieve better performance than TA.
>
> **Q2. The experimental setup on A100 should be further claimed.**
>
> A2: The requirement of 4 GPUs applies exclusively to the scenario that the parameter size of target model is 70B. We clarify that our experiments on the NVIDIA A100 GPU did not include the 70B variant. All A100-based evaluations were conducted with 7B/8B/13B-scale Target Models.
>
> **Q3. Compare and discuss Cerberus[1], DDD[2], Hass[3].**
>
> A3:
>
> **(1) Cerberus [1]:**
>
> The observation in Cerberus that earlier heads exhibit higher accuracy aligns with our understanding. This is because, in parallel decoding, later heads predict tokens farther from the input (e.g., predicting the $i+2$-th token based on the $i$-th hidden state introduces greater error compared to predicting the $i+1$-th token). However, Cerberus still employs identical parameter scales and architectures across all heads, failing to prioritize earlier heads—a key distinction from our method. Their primary contribution lies in introducing sequential knowledge for parallel heads, whereas our approach explicitly optimizes model structure to prioritize early tokens.
>
> **(2) DDD [2]:**
>
> DDD improves Eagle2 by using a dynamic draft tree. Since DDD’s code is not open-sourced, we compare relative improvements over Eagle2. On the MT-Bench dataset, the performance gains of DDD and Gumiho are:
>
> |model|DDD|Gumiho|
> |-|-|-|
> |V 7B|3.5%|**9.4%**|
> |V 13B|4.2%|**6.3%**|
> |L2 7B|2.3%|**5.5%**|
> |L2 13B|2.3%|**5.4%**|
>
> Our method consistently outperforms DDD across all tested models, demonstrating our effectiveness.
>
> **(3) Hass [3]:**
>
> Hass improves Eagle2 by addressing the training-inference inconsistency in serial transformer heads within SpD. While Hass focuses on training methodology, our work enhances performance via architectural modifications. These two directions are orthogonal: our method could integrate Hass’s training improvements for further gains.
>
>
> In the table below, we compare the performance of our Gumiho with Hass. Additionally, we evaluate a variant of our model (denoted as Gumiho w/ Hass ) where Hass’s training method is applied to our Transformer Head. All results are conducted on an Mi250 GPU using the LLaMA3-8B model with a temperature of 0.
>
> | | MT-Bench | HumanEval |
> |----|----|---|
> | Eagle2 | 2.16×| 2.51×|
> | Hass| 2.26×| 2.59×|
> | Gumiho| 2.38×| 2.77×|
> | Gumiho w/ Hass| **2.43×**| **2.84×**|
>
> The aforementioned discussions will be included in the final version of our paper.
>
> **Q4. Why $p_i - \zeta < 0$ ($i=d+1$) is not assumed and discussed accordingly.**
>
> A4: Your concern is intuitively reasonable. From the perspective of probability definitions, it would indeed seem necessary to assume that probabilities should be greater than 0, even if this auxiliary variable is not a real probability but merely a constructed parameter for our proof.
>
> However, throughout the proof process, we observed that such a constraint was unnecessary, and the proof could be completed without invoking this additional condition. Therefore, to maintain conciseness, we did not explicitly discuss this case separately.
>
> **Q5. In Appendix D, the authors show the comparison of Eagle2 and Gumiho on two datasets. What about the other four datasets?**
>
> A5: The other four datasets also exhibit similar results, and we will include the comparisons of these datasets in the final version of the manuscript.

---

> > ### Comment · Reviewer_E4Fx · 2025-04-03
> >
> > I am satisfied with the rebuttal from the author. They have addressed all of my concerns, and I decide to keep my decision.

---

> > > ### Author Response · Authors · 2025-04-07
> > >
> > > Thank you for your feedback and your support to our paper.

---

### Official Review · Reviewer_xLKt · 2025-03-14

**Overall Recommendation:** 3

**Summary:**

This paper proposes Gumiho, a hybrid architecture to prioritize early draft tokens with large and autoregressive heads, and parallel decoding for the rest tokens. The experimental results strongly support the effectiveness of this method.

**Claims And Evidence:**

The experimental results strongly support the effectiveness of this method. Not only does it achieve a speedup in wall time, it also increases the acceptance rate—validating the authors’ approach from two different angles.

**Essential References Not Discussed:**

no

**Experimental Designs Or Analyses:**

Yes.

But I think the authors should include more experimental results compared to the results of the implementation with flash decoding, i.e., flash_attn_with_kvcache and bsz > 1.

**Methods And Evaluation Criteria:**

Yes, LLM acceleration is a very important topic for current applications. However, the authors should test over more advanced settings like flash decoding and bsz > 1.

**Other Comments Or Suggestions:**

NA.

**Other Strengths And Weaknesses:**

Advantages:
1. The writing in this article is very clear.
2. I think the idea presented here makes a lot of sense. Intuitively, because the model is autoregressive, the earlier tokens are definitely more important than the later ones. Moreover, the proposed solution is quite elegant.
3. The experimental results strongly support the effectiveness of this method. Not only does it achieve a speedup in wall time, it also increases the acceptance rate—validating the authors’ approach from two different angles.
4. Balanced Performance and Latency: Gumiho’s design leverages a serial Transformer head to boost the quality of early token predictions without significantly sacrificing speed. The authors validate this balance through ablation studies (Figure 4), showing that the performance gains outweigh the latency introduced by the serial component.

Disadvantages:
1. The paper lacks a direct comparison with a strong baseline using flashdecoding.
2. The authors seem to introduce two different draft models (one large and one small), which could result in additional memory usage.
3. For the early draft model, using an approach similar to Eagle introduces an inconsistency between training and testing. I believe this issue becomes more pronounced as the network depth increases, as shown in Figure 4. Could the hass training method help mitigate this problem?
4. throughput improvement.

If the authors report results on 1 & 4, I will increase score from 2 to 3.

**Questions For Authors:**

see cons

**Relation To Broader Scientific Literature:**

na.

**Theoretical Claims:**

I checked the correctness of the theory and it is correct.

---

> ### Author Rebuttal · Authors · 2025-03-31
>
> Thank you for your suggestions and support.
>
> **Q1. The paper lacks a direct comparison with a strong baseline using flashdecoding.**
>
> A1: We have evaluated the baseline with FlashDecoding (FD) on the MT-Bench dataset using Mi250 GPU, as shown in the table below.
>
> ||L2 7B|L2 13B|
> |-|-|-|
> |FlashDecoding|2.11×|2.12×|
> |Gumiho|**3.07×**|**3.34×**|
>
> It is important to clarify that FD is an algorithm specifically optimized for attention computation. Its acceleration mechanism for LLMs is orthogonal to SpD – meaning these two approaches can be combined to achieve further acceleration beyond standalone SpD or FD implementations. The results of the combination will be shown in the final version of our paper due to the limited time of rebuttal.
>
> **Q2. The authors seem to introduce two different draft models (one large and one small), which could result in additional memory usage.**
>
> A2: Yes, you are correct. Our method does require more memory usage compared to Eagle2, and Medusa. Here’s why:
> - Eagle2 use one single-layer Transformer head and reuse it serially across multiple speculative steps.
> - Medusa only maintains five parallel MLP heads.
> - Our approach, however, employs a two-layer Transformer head alongside five parallel MLP heads.
>
> This architectural choice increases memory consumption. However, it enables a hybrid generation strategy: serial decoding for the first few tokens (to ensure accuracy) and parallel decoding for subsequent tokens (to maximize speed). While this introduces a memory-accuracy trade-off, it achieves a significant speedup improvement. We will explicitly discuss this trade-off in the final manuscript.
>
> **Q3. For the early draft model, using an approach similar to Eagle introduces an inconsistency between training and testing. I believe this issue becomes more pronounced as the network depth increases, as shown in Figure 4. Could the hass training method help mitigate this problem?**
>
> A3: Yes, our method could integrate Hass’s training strategy for further gains. Hass improves Eagle2 by addressing the training-inference inconsistency in serial transformer heads within SPD. While Hass focuses on training methodology, our work enhances performance via architectural modifications. These two directions are orthogonal, so our method could combine with Hass for further improvements.
>
> In the table below, we compare the performance of our Gumiho with Hass. Additionally, we evaluate a variant of our model (denoted as Gumiho w/ Hass ) where Hass’s training method is applied to our Transformer Head. All results are conducted on an Mi250 GPU using the LLaMA3-8B model with a temperature of 0.
>
> ||MT-Bench|HumanEval|
> |-|-|-|
> |Eagle2|2.16×|2.51×|
> |Hass|2.26×|2.59×|
> |Gumiho|2.38×|2.77×|
> |Gumiho w/ Hass|**2.43x**|**2.84x**|
>
> The aforementioned discussions will be included in the final version of our paper.
>
> **Q4. throughput improvement.**
>
> A4: We evaluated Gumiho under bs > 1 scenarios with Vicuna 7B, as shown in the table below. Experimental results demonstrate that the speedup effect degrades as batch size increases, aligning with observations in prior works like Eagle. Throughput speedup of baseline and Gumiho is computed at their respective maximum bs, as Eagle do.
>
> |BatchSize | MT-Bench | HumanEval |
> |-|-|-|
> | 1| 3.15×| 3.65×|
> | 2| 3.10×| 3.61×|
> | 4| 2.87×| 3.34×|
> | Throughput| 2.03x | 2.11x |
>
> The degradation in speedup with larger batch sizes can be primarily attributed to two key factors:
>
> **(1) Inter-Sequence Execution Time Imbalance.**
> With larger batch sizes, the number of accepted tokens varies across sequences within a batch, leading to divergent completion times. Due to the "bucket effect" in batch processing — where the entire batch's completion time is determined by the slowest sequence — completed sequences must wait for unfinished ones. This waiting time exhibits growth as batch size increases.
> Notably, directly removing completed sequences and inserting new ones theoretically mitigates this issue but proves impractical in reality. Newly inserted sequences require full context re-computation (prefill phase), whereas existing sequences leverage pre-cached KV for efficient decoding. This computational asymmetry further increases overall latency.
>
> **(2) Computational Resource Contention.**
> Large batch sizes push SpD into a computation-bound regime. During parallel verification, the target model must process a massive number of candidate tokens per step (quantity = draft tokens × batch size). Larger batch sizes prevent full parallelization of computing units, causing the verification process to degrade into partially sequential operations. This diminishes the theoretical acceleration benefits.
>
> Given these constraints, most SpD solutions (e.g., Lookahead, Medusa, Hydra, Cerberus, Eagle2, Hass) primarily focus on optimizing single-sequence (bs=1) scenario. Our work follows them, prioritizing inference efficiency optimization for single-sequence processing.

---

> > ### Comment · Reviewer_xLKt · 2025-04-09
> >
> > The results of hass are good, and thanks for the authors' rebuttal. I increased my score.

---

> > > ### Author Response · Authors · 2025-04-09
> > >
> > > Thank you for your feedback and your support to our paper.

---

### Decision · Program_Chairs · 2025-05-01

**Decision:**

Accept (poster)

**Comment:**

This paper proposes Gumiho, a hybrid architecture for speculative decoding in Large Language Models. The core idea is motivated by the insight that early tokens in a draft sequence are more critical for the overall acceptance rate than later ones. Gumiho operationalizes this by using a more accurate, serial Transformer-based head for early tokens and faster parallel MLP-based heads for later tokens.

Strengths highlighted by reviewers include the clear motivation, the intuitive and elegant hybrid design, the theoretical justification, and the comprehensive experiments demonstrating speedup improvements over strong baselines (like Eagle2) across various models and benchmarks (please add the additional results from the rebuttal). The paper is generally well-written and sound (please update the paper to address all the points discussed below to provide further details and clarification on the architecture and experimental setup).